# Quantifying the contributions of riverine vs. oceanic nitrogen to hypoxia in the East China Sea

Fabian Große[1,2], Katja Fennel[1], Haiyan Zhang[1,3], and Arnaud Laurent[1]

[1]Department of Oceanography, Dalhousie University, Halifax, NS, Canada
[2]Department of Mathematics and Statistics, University of Strathclyde, Glasgow, United Kingdom
[3]School of Marine Science and Technology, Tianjin University, Tianjin, China

**Correspondence:** Fabian Große (fabian.grosse@dal.ca)

**Abstract.** In the East China Sea, hypoxia (oxygen $\leq 62.5\,\mathrm{mmol\,m^{-3}}$) is frequently observed off the Changjiang (or Yangtze) River estuary covering up to about $15{,}000\,\mathrm{km^2}$. The Changjiang River is a major contributor to hypoxia formation because it discharges large amounts of freshwater and nutrients into the region. However, modelling and observational studies have suggested that intrusions of nutrient-rich oceanic water from the Kuroshio Current also contribute to hypoxia formation. The relative contributions of riverine versus oceanic nutrient sources to hypoxia have not been estimated before. Here, we combine a three-dimensional, physical-biogeochemical model with an element tracing method to quantify the relative contributions of nitrogen from different riverine and oceanic sources to hypoxia formation during 2008–2013. Our results suggest that the hypoxic region north of $30\,°\mathrm{N}$ is dominated by Changjiang River inputs, with its nitrogen loads supporting 74% of oxygen consumption. South of $30\,°\mathrm{N}$, oceanic nitrogen sources become more important supporting 39% of oxygen consumption during the hypoxic season, but the Changjiang River remains the main control of hypoxia formation also in this region. Model scenarios with reduced Changjiang River nitrogen loads and reduced open-ocean oxygen levels suggest that nitrogen load reductions can significantly reduce hypoxia in the East China Sea and counteract a potential future decline in oxygen supply from the open ocean into the region.

## 1 Introduction

In the East China Sea (ECS), hypoxic conditions (i.e. dissolved oxygen ($O_2$) concentrations $\leq 62.5\,\mathrm{mmol\,m^{-3}}$) are frequently observed off the Changjiang River (or Yangtze) Estuary covering up to about $15{,}000\,\mathrm{km^2}$ (Li et al., 2002; Zhu et al., 2017). Hypoxia was first reported in 1959 (Zhu et al., 2011), and a significant increase in its spatial extent has been observed since the 1980s (Li et al., 2011; Wang, 2009; Wang et al., 2015; Zhu et al., 2011). The Changjiang River is the fifth largest river in the world in terms of freshwater (FW) discharge ($9 \times 10^{11}\,\mathrm{m^3\,y^{-1}}$) and its nutrient concentrations are comparable to other strongly anthropogenically affected rivers (Liu et al., 2003). The observed increase in hypoxic area since the 1980s has been

attributed to elevated nutrient loads due to fertilizer use in the Changjiang watershed (Siswanto et al., 2008; Wu et al., 2019; Yan et al., 2003).

Various studies have suggested that oceanic nutrients also play a role in hypoxia formation in the ECS (Chi et al., 2017;
Li et al., 2002; Wang et al., 2018; Zhou et al., 2017b, 2018; Zhu et al., 2011). The importance of oceanic nutrient supply distinguishes hypoxia in the ECS from the otherwise comparable situation in the northern Gulf of Mexico (NGoM), where a similar spatial extent of hypoxic conditions is fueled by freshwater and anthropogenic nutrient inputs from a major river, the Mississippi (Fennel and Testa, 2019). Observations in the ECS indicate that south of 30 °N, intrusions of nutrient-rich water from the Kuroshio Current influence the shelf dynamics (Wang et al., 2018; Zhou et al., 2017b, 2018). These intrusions vary
seasonally, with stronger intrusions in winter (Bian et al., 2013; Guo et al., 2006). However, northward water mass transport on the ECS shelf is stronger in summer than in winter (Guo et al., 2006), supported by the weak northeastward winds during the East Asian summer monsoon.

The complexity of the circulation and importance of different nutrient sources for hypoxia development make this system particularly amenable to model analyses with high spatio-temporal resolution (Fan and Song, 2014; Zhao and Guo, 2011;
Zhang et al., same issue; Zheng et al., 2016; Zhou et al., 2017a; Zhang et al., 2018). Fan and Song (2014) used simulated salinity and nutrient distributions to show that Changjiang River inputs are transported southward in winter and northward in summer. Zhao and Guo (2011) and Zhou et al. (2017a) used model sensitivity studies to highlight the role of oceanic nutrient sources on productivity and hypoxia in the ECS, respectively. Zhang et al. (2019) combined a physical-biogeochemical model with an element tracing method (e.g. Ménesguen et al., 2006) to quantify the relative contributions of different nutrient sources
to primary production (PP). They found that riverine nitrogen (N) supports 56% of water column integrated PP in the ECS regions shallower than 50 m. With organic matter degradation being the main sink of $O_2$ in the subsurface waters of the ECS (Li et al., 2002), this suggests that riverine N also dominates $O_2$ consumption. However, a quantification of the relative contributions of riverine versus oceanic nutrient sources to hypoxia has not been available until now.

By combining a high-resolution biogeochemical model with this active element tracing method expanded for the quantifica-
tion of the contributions to $O_2$ processes (Große et al., 2017, 2019), we provide such an analysis here. We apply the element tracing method to an implementation of the Regional Ocean Modeling System (ROMS; Fennel et al., 2006; Haidvogel et al., 2008) configured for the ECS (Zhang et al., same issue). This allows us to quantify the contributions of N from different riverine and oceanic sources to hypoxia formation in the ECS and analyze year-to-year and seasonal variability in the individual contributions resulting from the East Asian monsoon cycle. In addition to supplying nutrients, the intrusions of open-ocean
subsurface waters are relevant to hypoxia by preconditioning $O_2$ concentrations in the region. Subsurface $O_2$ has declined over the past decades in the northwest Pacific (Schmidtko et al., 2017) and is projected to continue decreasing in the future (Bopp et al., 2017), which may further exacerbate hypoxia in the region (Qian et al., 2017). We analyze the effect of reduced open-ocean $O_2$ concentrations on hypoxia under current and reduced N loads from the Changjiang River and compare our results to previous findings for the NGoM's hypoxic zone.

## 2  Methods

### 2.1  The physical-biogeochemical model

We used an implementation of ROMS (Haidvogel et al., 2008) for the ECS (Bian et al., 2013). The model covers the region from 116 °E to 134 °E and from 20 °N to 42 °N with a resolution of 1/12 ° (Fig. 1) with 30 terrain-following $\sigma$-layers.

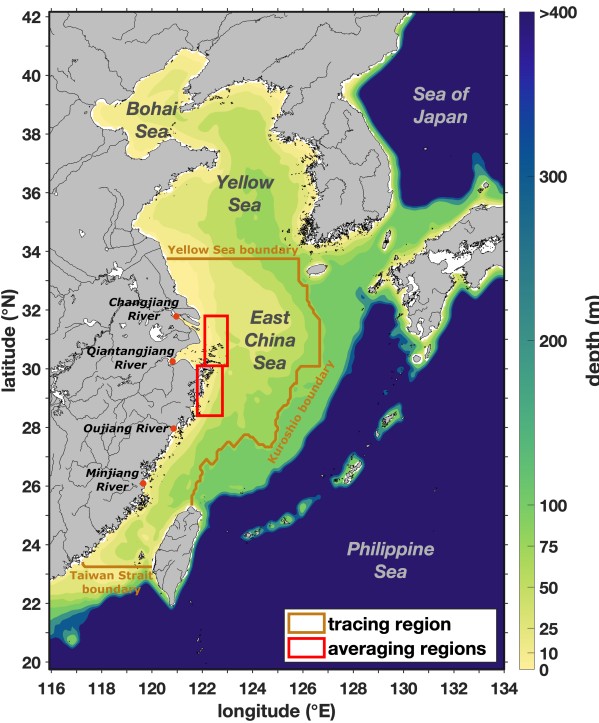

**Figure 1.** Model domain and bathymetry, with sub-domain used for nitrogen tracing, rivers inside the tracing region, and northern and southern regions used for time series analysis. © The GMT Team, 2018.

The biogeochemical component is based on the N-cycle model of Fennel et al. (2006, 2011) but was expanded to include phosphate Laurent et al. (2012), oxygen (Fennel et al., 2013) and riverine dissolved organic matter (Yu et al., 2015). The model state variables are: nitrate ($NO_3^-$), ammonium ($NH_4^+$), phosphate ($PO_4^{3-}$), one functional group each for phyto- and zooplankton, small and large detritus, riverine dissolved organic matter, and dissolved $O_2$. Riverine dissolved organic matter is explicitly represented to account for the more refractory nature of river-borne organic matter compared to organic matter produced in the marine environment. This is also reflected in the one order of magnitude lower remineralization rate compared to small detritus (Yu et al., 2015). Instantaneous benthic remineralization was applied at the sediment-water interface (Fennel et al., 2006, 2011). This implies that all organic matter that sinks to the seafloor is remineralized immediately, with 75% of the deposited N being lost to dinitrogen via benthic denitrification (Fennel et al., 2006). Sediment $O_2$ consumption is calculated

from the benthic remineralization flux multiplied by a molar ratio of $\sim$115:4 between $O_2$ uptake and release of $NH_4^+$ (Fennel et al., 2013). Light attenuation was expanded by including a term dependent on bottom depth (Zhang et al., same issue). For a

complete set of model equations, we refer the reader to the Appendix of Laurent et al. (2017).

Initial and open boundary conditions for temperature and salinity were derived from World Ocean Atlas 2013 version 2 (WOA13-v2) climatologies (Locarnini et al., 2013; Zweng et al., 2013). Temperature and salinity were nudged weekly toward the climatology using a nudging scale of 120 days. Horizontal velocities and sea surface elevation at the open boundaries are based on the SODA reanalysis (Carton and Giese, 2008). Eight tidal constituents ($M_2$, $S_2$, $N_2$, $K_2$, $K_1$, $O_1$, $P_1$ and $Q_1$) are

imposed using tidal elevations, and tidal currents are derived from the global tide model of Egbert and Erofeeva (2002). Initial and open boundary conditions for $NO_3^-$, $PO_4^{3-}$ and $O_2$ are also based on WOA13-v2 (Garcia et al., 2013a, b), while small positive values are prescribed for all other biogeochemical variables. In regions deeper than 100 m and in the Yellow Sea north of 34 °N, $NO_3^-$ concentrations are nudged toward the climatology using time constants of 7 and 10 days, respectively. The results are not sensitive to the exact choice of the nudging time scale and, in fact, are almost indistinguishable if the times

scales are varied between one week and two weeks.

The model was run for the period 2006–2013, with the first two years used as spin-up, and forced by 6-hourly wind stress, surface heat and FW fluxes from the ECMWF ERA-Interim dataset (Dee et al., 2011). Daily FW discharge and nutrient loads for 11 rivers were imposed (see supplement Table S1 for river locations), with the Changjiang River being by far the largest. Discharge for the Changjiang is obtained from Datong Hydrological Station (http://www.cjh.com.cn/en/). Concentrations of

$NO_3^-$, $NH_4^+$ and $PO_4^{3-}$ for the Changjiang River were obtained from the monthly Global NEWS data set (Seitzinger et al., 2005). For the other rivers, FW discharge and nutrient loads were prescribed using climatologies (Liu et al., 2009; Tong et al., 2015; Zhang, 1996). Due to the lack of data on organic matter loads, river load concentrations of small and large detritus and dissolved organic N were assumed conservatively at $0.5\,\mathrm{mmol\,N\,m^{-3}}$, $0.2\,\mathrm{mmol\,N\,m^{-3}}$ and $15\,\mathrm{mmol\,N\,m^{-3}}$, respectively. Riverine concentrations for phyto- and zooplankton were assumed equal to their pelagic concentrations. The

model demonstrates good skill in representing the hydrography and biogeochemistry of the ECS (Zhang et al., same issue; Figs. S2–S4) and reproduces the main features of the ECS circulation (Fig. S1). The simulation using this setup is hereafter referred to as the reference simulation.

In addition to the reference simulation, three scenario simulations were performed: First, to assess the influence of a reduction in riverine N load on hypoxia, a nutrient reduction scenario was run with 50% smaller N concentrations in the Changjiang River

input. All other forcing remained the same. The reduction of riverine N only reflects the assumption that nutrient reductions are achieved primarily by reduced application of industrial fertilizer in agriculture–the main source of excess N loads to the ECS (Yan et al., 2003). Second, to investigate the potential effect of reduced open-ocean $O_2$ supply, a scenario similar to the reference simulation but with reduced $O_2$ in the open ocean was performed where the initial and open-boundary $O_2$ concentrations were reduced by 20% throughout the water column in regions deeper than 200 m. This implies that simulated

$O_2$ transport into the model domain is reduced by 20% relative to the reference simulation, which corresponds to the changes projected by Earth System Models under an RCP8.5 scenario for the Northwest Pacific Ocean at the end of the 21st century (Bopp et al., 2017). Third, the reduced open-ocean $O_2$ scenario was repeated for reduced river N as in the N reduction scenario.

## 2.2 Passive freshwater and active nitrogen tracing

Passive dye tracers were used to track FW inputs from the Changjiang River similar to previous ROMS applications (Große et al., 2019; Hetland and Zhang, 2017; Rutherford and Fennel, 2018; Zhang et al., 2010, 2012). The rate of change of the concentration of a passive dye tracer from the $i$-th source ($C_p^i$) is described as:

$$\frac{\partial C_p^i}{\partial t} = \nabla \cdot \left( \overline{\overline{D}} \nabla C_p^i \right) - \nabla \cdot \left( C_p^i \boldsymbol{v} \right) + S_{C_p^i}. \tag{1}$$

Here, $\overline{\overline{D}}$ is the second-order diffusion tensor (or diffusivity), $\boldsymbol{v}$ is the velocity vector, and $S_{C_p^i}$ represents the external dye tracer sources (i.e. riverine FW discharge). The dye tracer was initialized with zero in the entire model domain. Changjiang FW discharge had a dye tracer concentration of 1 and the dye tracer was used to analyze the influence of FW from the Changjiang River on stratification (i.e. potential energy anomaly; Simpson, 1981).

In addition, we applied an active element tracing method (Große et al., 2017; Ménesguen et al., 2006; Radtke et al., 2012) to quantify the contributions of different N sources to $O_2$ consumption. For this purpose, each model variable containing N is subdivided into fractions from the different source regions or rivers. The rate of change of a specific source fraction is described as:

$$\frac{\partial C_X^i}{\partial t} = \nabla \cdot \left( \overline{\overline{D}} \nabla C_X^i \right) - \nabla \cdot \left( C_X^i \boldsymbol{v} \right) + R_{C_X} \cdot \frac{C_{X_{con}}^i}{C_{X_{con}}}. \tag{2}$$

$C_X$ and $C_X^i$ represent the concentrations of state variable $X$ (e.g. $NO_3^-$) and its labeled fraction from the $i$-th source (e.g. $NO_3^-$ from the Changjiang River), respectively. $R_{C_X}$ describes internal and external sources and sinks of $X$. The index '$con$' in the source/sink term implies that the relative concentration of the variable consumed by a process is used; e.g. source-specific nitrification is calculated based on the relative concentration of $NH_4^+$. In our study, Eq. (2) is solved diagnostically for N tracers from each traced source using the daily model output for the complete N cycle and the post-processing software of Große et al. (2017), which has been adapted for ROMS (Große et al., 2019). Essentially, a source-specific flux over a calculation time step is the product of the 'bulk' flux (e.g. total $NO_3^-$ uptake during PP) with the relative fraction of the source-specific state variable (e.g. $NO_3^-$ containing N from the Changjiang River) divided by its corresponding 'bulk' state variable (e.g. $NO_3^-$).

N tracing is only applied in the region without $NO_3^-$ nudging (see 'tracing region' in Fig. 1). Inside this region, we simultaneously traced N from five different sources: the Changjiang River, three other smaller rivers (Minjiang, Oujiang and Qiantangjiang Rivers; grouped into one source; see Table S1), the Taiwan Strait (at the southern boundary of the tracing region), the Kuroshio Current (at the eastern boundary), and the Yellow Sea (at the northern boundary; see Fig. 1). As all N tracers that enter the tracing region across the Taiwan Strait, Kuroshio or Yellow Sea boundaries are labeled as such, this implies that N tracers leaving the tracing region cannot reenter. This tracer setup is similar to that of Zhang et al. (2019), with the difference that we separate the Changjiang River from smaller rivers and that we do not account for atmospheric nutrient deposition.

We applied the N tracing to the reference simulation. Since the initial distributions of N tracers from the different sources are not known, we apply a spin-up procedure to the tracing, which provides the initial distributions of the labeled N tracers at the beginning of the analysis period 2008–2013. We first re-ran year 2006 three times. For the first iteration, all N mass already in

the system was arbitrarily attributed to the small rivers, while subsequent iterations were initialized from the final distribution of the previous iteration. Comparison of the final states of two subsequent iterations confirmed that a dynamic steady state of the labeled N tracer distributions was reached after three iterations. We then ran year 2007 as an additional spin-up year to ensure that the spatial distributions at the beginning of 2008 are not affected by re-running year 2006.

## 3 Results

### 3.1 Changjiang freshwater discharge and nitrogen concentrations


The Changjiang River is the main source of FW and nutrients in the ECS. Daily time series of its FW discharge and total nitrogen (TN) concentrations (i.e. the sum of the riverine concentrations of all N variables: $NO_3^-$, $NH_4^+$, small and large detritus, dissolved organic matter, phyto- and zooplankton) are shown in Fig. 2. The FW discharge has a distinct seasonal cycle (highest in summer, lowest in winter) due to the monsoon season with high precipitation in summer (Wang et al., 2008),

and significant year-to-year variability. The TN concentrations also show high year-to-year variability. TN concentrations are dominated by $NO_3^-$ concentrations, with detritus and dissolved organic N contributing only 8—13%.

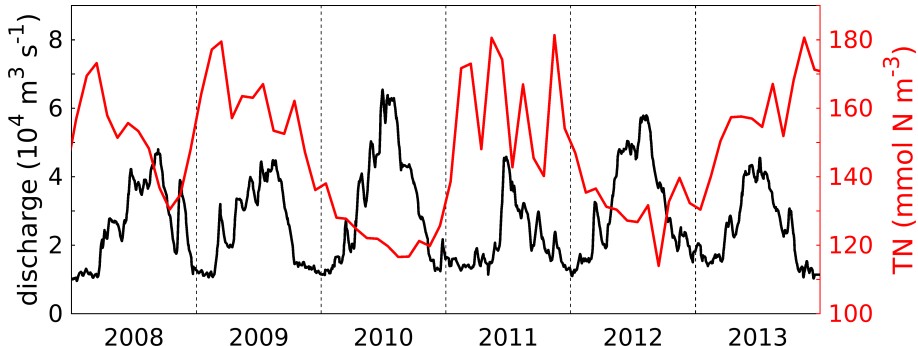

**Figure 2.** Time series of daily freshwater discharge (black, left y-axis) and monthly total nitrogen (TN) concentration (i.e. the sum of the riverine concentrations of all N variables: $NO_3^-$, $NH_4^+$, small and large detritus, dissolved organic matter, phyto- and zooplankton) in the Changjiang River (red, right y-axis).

### 3.2 Spatial patterns in oxygen consumption

We focus our analysis of $O_2$ dynamics on gross $O_2$ consumption (GOC), i.e. the sum of sediment $O_2$ consumption (SOC) and water column respiration (WR; incl. nitrification), and integrate over the 12 deepest pelagic layers (analogous to Zhang

et al., same issue). This is reasonable as Zhang et al. show that both SOC and WR are relevant $O_2$ sinks in the hypoxic bottom boundary layer of the ECS. Hypoxia in the ECS is most pronounced between July and November; we hence focus most of our

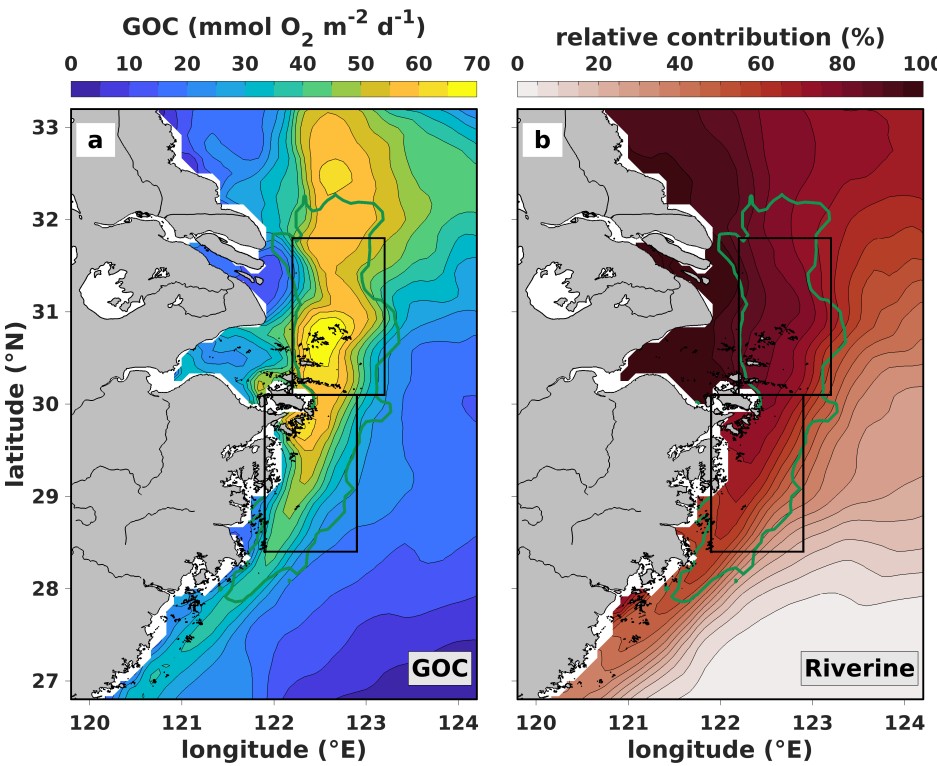

**Figure 3.** (a) Gross $O_2$ consumption (GOC) and (b) relative contribution supported by nitrogen from rivers (Changjiang and other rivers) averaged from July to November 2008–2013. Green line: area affected by hypoxia. Boxes: analysis regions used in sections 3.3 and 3.4.

analyses on this time period. First, we analyze the general spatial patterns in GOC and the relative contributions from riverine and oceanic N sources.

Figures 3a and b show maps of GOC and its relative riverine contribution averaged from July to November over the years 2008 to 2013, respectively. GOC is highest (up to $70\,\mathrm{mmol\,O_2\,m^{-2}\,d^{-1}}$) in the northern part (30 °N to 31 °N) of the region typically affected by hypoxia (indicated by the green line). Farther north, GOC is still high (50 to 60 $\mathrm{mmol\,O_2\,m^{-2}\,d^{-1}}$), but decreases significantly in the offshore and southward directions with the strongest gradient roughly between 29 °, 122.5 °E and 31 °N, 123 °E. The riverine contribution to GOC is highest (>95%) in the coastal regions between 30 °N and 32 °N and steadily decreases offshore and southward. South of 32 °N, the strongest gradient in the riverine contribution corresponds to the maximum gradient in GOC.

Based on the hypoxic area locations simulated in our model and the spatial patterns in GOC and its riverine contribution, we defined two distinct analysis regions (black boxes in Fig. 3): a northern region where GOC is supported mainly by riverine N (>70%) and a southern region where the riverine contribution declines strongly from 75% to 20% in southeastward direction.

These regions are used to quantify how the relative contributions of riverine (Changjiang River and smaller rivers) and oceanic N sources (Kuroshio, Taiwan Strait and Yellow Sea) to GOC differ regionally.

## 3.3 Year-to-year variability in source-specific oxygen consumption

In order to provide insight in the relative importance of the different N sources for hypoxia formation in the northern and southern hypoxic regions, Fig. 4 shows time series of average source-specific GOC and total hypoxic area from July to November for 2008 to 2013 in both regions. Here, we define total hypoxic area as the area experiencing hypoxia at any time during July to November. The corresponding values of total GOC and the relative contributions of the different sources are provided in Table S2.

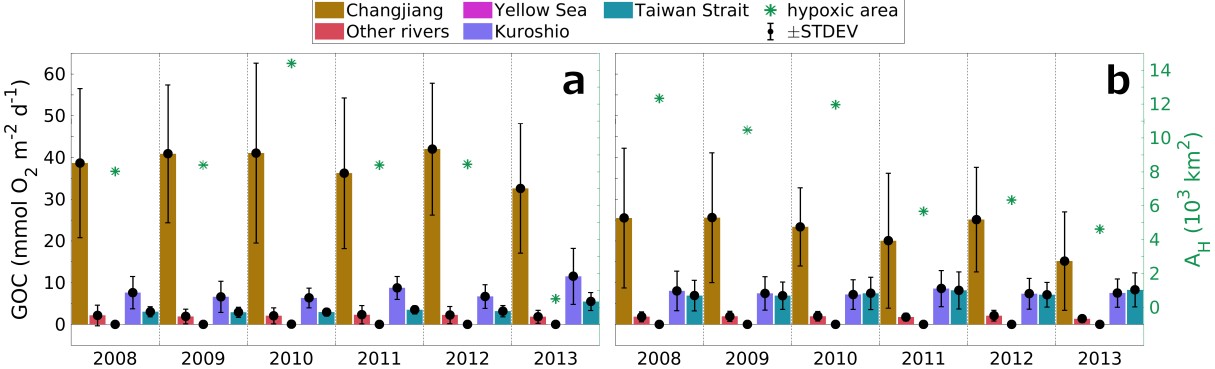

**Figure 4.** Average source-specific gross $O_2$ consumption (GOC) and total hypoxic area during July–November of 2008 to 2013 in the (a) northern and (b) southern regions (see Fig. 1). Same legend for both panels.

In the northern region (Fig. 4a), riverine N sources (Changjiang and other rivers) account for $78.0 \pm 5.9\%$ of GOC averaged over 2008–2013, while oceanic sources support only $22.0 \pm 5.9\%$. The Changjiang River constitutes the largest contribution ranging between 63.3% in 2013 and 78.2% in 2010, which are also the years of the smallest and largest hypoxic areas in this time series, respectively. This indicates that the Changjiang River is the main control on $O_2$ consumption in the northern region.

In the southern region (Fig. 4b), total GOC is about 24% lower than in the northern region, and the riverine contribution is also lower ($61.6 \pm 5.7\%$). The average Changjiang contribution is $56.9 \pm 5.1\%$, while the contributions from the Kuroshio and Taiwan Strait account for $19.5 \pm 2.2\%$ and $18.9 \pm 3.2\%$, respectively. Clearly, N sources other than from the rivers play a role in the southern region.

Interestingly, larger hypoxic areas tend to coincide with high contributions from the Changjiang River and small oceanic contributions (2008–2010), while the tends to be Changjiang is less important in years of small hypoxic areas (2011 and 2013). This suggests that the water mass distribution is an important factor for controlling the extent of hypoxia in the southern region, with a higher Changjiang contribution supporting larger hypoxic areas.

The water mass distribution in the southern region is strongly influenced by large-scale wind patterns, i.e. the East Asian monsoon, and variations in the wind field may affect both seasonal and year-to-year variability in GOC and thus hypoxia. Next, we analyze the seasonality of source-specific GOC and hypoxic area in relation to the large-scale winds in the southern region in years of small and large hypoxic areas.

### 3.4  Seasonal cycle of oxygen consumption and hypoxia in the southern region

In the southern region, the largest and smallest hypoxic areas are simulated in 2008 and 2013, respectively. Figure 5 presents monthly time series of source-specific GOC and total hypoxic area for both years. In addition, it shows the 6-year monthly average of the meridional wind speed 10 m above sea level ($v_{10}$) and the corresponding anomalies for both years.

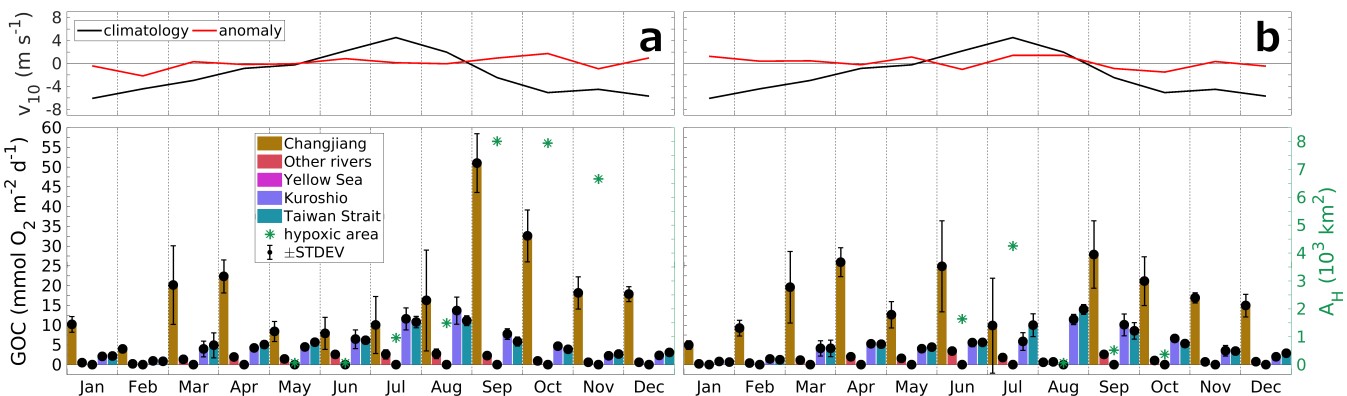

**Figure 5.** Monthly time series of source-specific contributions to GOC and total hypoxic area ($A_H$) in the southern region (see Fig. 1) in (a) 2008 (year of largest $A_H$) and (b) 2013 (smallest $A_H$), and anomaly of northward wind speed 10 m above sea level ($v_{10}$) relative to 2008–2013 ('climatology') averaged over the ECS (25–33 °N, 119–125 °E). Same legend and axes for both panels.

In general, the Changjiang's GOC fraction tends to be high in spring (March and April) and in late summer (September). This seasonal pattern is explained by shifts in the dominant wind direction (Yang et al., 2012), where the transition from southward to northward winds from March to July supports a northward transport of both the Changjiang River plume and Kuroshio and Taiwan Strait waters increasing the oceanic contribution to GOC. In contrast, the reversal from northward to southward winds in September results in a southward movement of river-influenced water masses. Year-to-year differences in hypoxia, e.g. in June or September/October, result from differences in the Changjiang contribution. The large Changjiang contribution to GOC in September/October 2008 (Fig. 5a) and in June 2013 (Fig. 5b) are followed by large hypoxic areas, while hypoxia almost vanishes in August 2013 when the Changjiang contribution diminishes. These increases (decreases) in the Changjiang contribution also coincide with significant increases (decreases) in FW thickness and thus stratification (see Fig. S5).

The significant year-to-year differences in both the Changjiang contribution to GOC and thus the hypoxic area can be related to year-to-year variability in the wind field and also in FW discharge. The anomalously weak southward winds in September/October of 2008 resulted in a weaker southward coastal current (see Fig. S6). This allowed for a longer presence

of Changjiang FW and nutrients in the region stimulating organic matter production and stabilizing vertical density stratifi-cation (see Fig. S5). The higher FW discharge in 2008 compared to 2013 (see Fig. 2) also supported stronger stratification. Consequently, the Changjiang contribution to GOC and thus hypoxic area remained large through October and only dropped in November. In June 2013, the anomalously weak northward winds allowed Changjiang water to be transported into the region (see Fig. S6). This caused an increase in GOC and – supported by less wind-induced mixing – stratification, and thus hypoxia. The opposite occurred in July/August of 2013, when anomalously strong northward winds pushed the Changjiang River water northward resulting in a decrease in GOC, stratification and thus hypoxia.

The results from sections 3.3 and 3.4 highlight the importance of Changjiang River nutrients for hypoxia formation in both the northern and the southern regions and suggest that N load reductions in the Changjiang River would mitigate hypoxia also in the southern region. This raises the question how a potential future decline in open-ocean $O_2$ concentrations would affect hypoxia in the ECS, which is analyzed next.

## 3.5   Potential future changes in hypoxia

The open-ocean water masses travelling to the hypoxic region are not only relevant as N sources but may also act as sources of low-$O_2$ water, thus preconditioning the region for hypoxia. Consequently, a future decline in subsurface open-ocean $O_2$ concentrations may exacerbate hypoxia in the ECS. To investigate how reduced open-ocean $O_2$ concentrations and reduced Changjiang River N loads would affect hypoxia, we performed the scenario simulations described in section 2.1. Figure 6 shows the cumulative hypoxic exposure in the bottom layer from July to November averaged over 2008–2013 for the reference simulation (Fig. 6a), the nutrient reduction scenario with 50% lower Changjiang River N loads (Fig. 6b), the scenario with 20% lower open-ocean $O_2$ concentrations (Fig. 6c) and the combined scenario with reduced N load and lower open-ocean $O_2$ (Fig. 6d). Table 1 presents the corresponding changes in different hypoxia metrics between the reference case and the scenarios.

In the reference case (Fig. 6a), hypoxia occurs between 28–32 °N and from the coast to about 123 °E, on average affecting $19.3 \pm 8.1 \times 10^3 \, \mathrm{km}^2$. Maximum hypoxic exposure is 28 days in the central parts of the hypoxic region. Hypoxia is still present in the simulation with 50% lower Changjiang River N loads (Fig. 6b), but its areal extent and hypoxic exposure are significantly reduced. Areas affected by continuous hypoxia of more than one and two weeks are reduced by >60%. Figure S7 shows that the 50% N load reduction in the Changjiang River significantly reduces the Changjiang contribution to GOC and the hypoxic area in the southern analysis region throughout the seasonal cycles of 2008 and 2013 (analogous to Fig. 5). Hypoxia vanishes entirely in most months with small hypoxic areas under current N loads (May to August 2008, September/October 2013), suggesting a relevant role of riverine N load reductions for hypoxia mitigation.

Under reduced open-ocean $O_2$ concentrations (Fig. 6c), hypoxic area increases by about 50% and regions affected by continu-ous hypoxia for more than one and two weeks expand by 86% and 118%, respectively (Table 1). Under reduced open-ocean $O_2$ and reduced Changjiang River N load (Fig. 6d), simulated changes relative to the reference are still negative, although comparably small (see Table 1). This suggests that N load reductions constitute a potent means to mitigate hypoxia under present conditions and to counteract deterioration of $O_2$ conditions in ECS due to open-ocean deoxygenation.

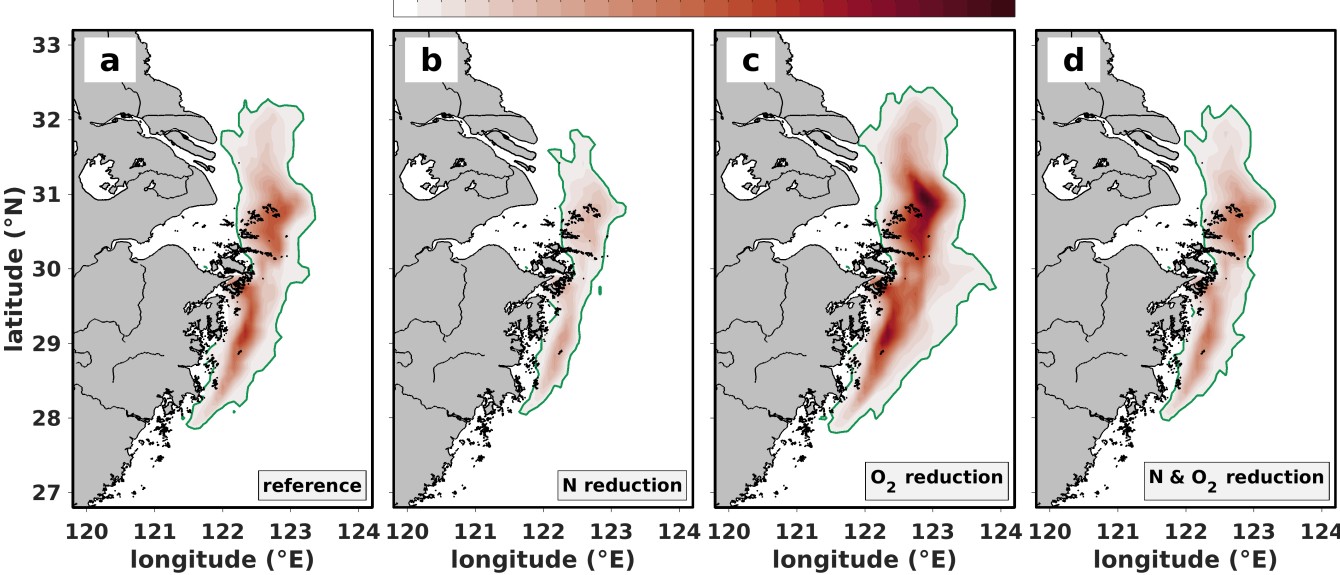

**Figure 6.** Average cumulative hypoxia during 2008–2013 derived from simulated bottom $O_2$ concentrations for (a) the reference simulation, (b) the nitrogen reduction scenario, (c) the reference simulation with reduced oceanic $O_2$ concentrations, and (d) the nitrogen reduction scenario with reduced oceanic $O_2$ concentrations.

## 4 Discussion

### 4.1 Model validation and study limitations

The here applied ROMS model demonstrates good skill in reproducing the hydrography and general circulation of the ECS
(Zhang et al., same issue; Fig. S1) and agrees well with both observations (Zhou et al., 2017b, 2018) and other modeling studies (Bian et al., 2013; Guo et al., 2006; Yang et al., 2011, 2012).

The simulated spatial distributions of dissolved inorganic nitrogen (DIN) are in agreement with observations of Gao et al. (2015) (see Figs. S2-S4). This, together with the good representation of the circulation, is essential for reliable results of the N tracing. Monthly averaged simulated SOC rates in the analysis regions result in 0.3–69.8 $\mathrm{mmol\,O_2\,m^{-2}\,d^{-1}}$ and are usually
within the observed range of 1.7–62.5 $\mathrm{mmol\,O_2\,m^{-2}\,d^{-1}}$ (Song et al., 2016; Zhang et al., 2017). Simulated rates higher than the observed ones only occur in July 2008 and August 2010, which is also the time of year of highest observed rates (Zhang et al., 2017). Zhang et al. (same issue) further show that simulated (daily) hypoxic areas are in agreement with observations, which range between 2,500 $\mathrm{km^2}$ and 15,000 $\mathrm{km^2}$ (Zhu et al., 2017). It should be noted that our values reported for the reference simulation in Table 1 exceed these values as they state the total area affected by hypoxia during each year, while observations
are limited to individual months.

**Table 1.** Hypoxia metrics for the four different cases presented in Fig. 6. $A_H$: hypoxic area; $A_{H,1w}$/$A_{H,2w}$: areas with continuous hypoxia for one and two weeks, respectively; $A_A$: anoxic area (i.e. sum of areas of all grid cells with at least one day of $O_2 = 0\,\mathrm{mmol\,m^{-3}}$; all in $10^3\,\mathrm{km^2}$). Values for the reference simulation are total areas. Other values are changes in area ($\Delta$) relative to the reference simulation.

| Metric | 2008 | 2009 | 2010 | 2011 | 2012 | 2013 | mean $\pm$ stdev |
|---|---|---|---|---|---|---|---|
| | | | | Reference | | | |
| $A_H$ | 23.3 | 20.8 | 31.0 | 16.3 | 18.2 | 6.4 | $19.3 \pm 8.1$ |
| $A_{H,1w}$ | 11.2 | 11.5 | 18.7 | 2.8 | 5.2 | 0.1 | $8.2 \pm 6.8$ |
| $A_{H,2w}$ | 4.0 | 5.7 | 11.8 | 0.6 | 0.9 | 0.0 | $3.8 \pm 4.5$ |
| $A_A$ | 1.8 | 2.2 | 3.5 | 0.1 | 0.6 | 0.0 | $1.4 \pm 1.4$ |
| | | | | N reduction | | | |
| $\Delta A_H$ | −11.2 | −8.3 | −12.4 | −14.9 | −13.8 | −4.9 | $-10.9 \pm 3.8$ |
| $\Delta A_{H,1w}$ | −6.9 | −7.4 | −11.7 | −2.8 | −4.7 | −0.1 | $-5.6 \pm 4.0$ |
| $\Delta A_{H,2w}$ | −3.0 | −4.4 | −8.8 | −0.6 | −0.9 | 0.0 | $-2.9 \pm 3.3$ |
| $\Delta A_A$ | −1.3 | −1.9 | −3.5 | −0.1 | −0.6 | 0.0 | $-1.4 \pm 1.3$ |
| | | | | $O_2$ reduction | | | |
| $\Delta A_H$ | 13.6 | 10.7 | 9.3 | 10.6 | 11.4 | 9.3 | $10.8 \pm 1.6$ |
| $\Delta A_{H,1w}$ | 10.4 | 6.6 | 7.5 | 6.5 | 9.3 | 2.3 | $7.1 \pm 5.1$ |
| $\Delta A_{H,2w}$ | 6.8 | 6.8 | 7.0 | 3.0 | 3.1 | 0.0 | $4.5 \pm 2.9$ |
| $\Delta A_A$ | 0.5 | 0.5 | 2.8 | 0.1 | 0.3 | 0.0 | $0.7 \pm 1.1$ |
| | | | | N and $O_2$ reduction | | | |
| $\Delta A_H$ | −2.3 | −2.5 | −3.9 | −11.8 | −5.8 | −3.8 | $-5.0 \pm 3.6$ |
| $\Delta A_{H,1w}$ | −2.1 | −2.5 | −4.4 | −2.8 | −3.2 | −0.1 | $-2.5 \pm 1.4$ |
| $\Delta A_{H,2w}$ | −1.1 | −1.6 | −4.1 | −0.6 | 0.0 | 0.0 | $-1.4 \pm 1.4$ |
| $\Delta A_A$ | −0.7 | −1.5 | −3.2 | −0.1 | 0.0 | 0.0 | $-1.0 \pm 1.2$ |

The instantaneous benthic remineralization (Fennel et al., 2006) used in our model does not take into account sediment burial. Song et al. (2016) estimated that averaged over the ECS shelf about 45% of deposited organic matter (ca. 14% of total primary production) is buried permanently in sediments, although with high spatial variability. This partly explains the slight overestimation of SOC rates. However, we consider it having only a small effect on the relative contributions of individual sources to GOC, as this limitation equally applies to all labeled N sources.

Shi et al. (2011) showed that the amount of suspended matter is significant in the coastal regions of the ECS. Sediment resuspension is not taken into account in this study. Its inclusion may result in a lower riverine contribution near the river mouths

and higher contributions in more distant areas (vice versa for oceanic contributions). Therefore, we recommend considering sediment resuspension in future studies.

It should further be noted that the $NO_3^-$ nudging to a climatology in the off-shelf areas and in the Yellow Sea does not resolve interannual variability in open-ocean $NO_3^-$ concentrations; however, we believe that these are small. Nutrient supply from the Kuroshio occurs primarily in the subsurface. With open-ocean subsurface concentrations showing significantly lower absolute values (e.g. Liu et al., 2016) and lower variability than coastal waters that are directly influenced by river inputs, neglecting interannual variations in open-ocean $NO_3^-$ concentrations seems justifiable. Variations in volume transport of Kuroshio in-

trusions are likely the main cause for interannual variations in N supply from the Kuroshio and are resolved by the model. Similarly, N concentrations in Taiwan Strait (e.g. Chen et al., 2004) are significantly lower than in the river inputs (by factor 10 to 100).

The diagnostic calculation of the source-specific N fluxes as the product of the fluxes for the non-labeled N tracers and the ratios of the labeled, source-specific tracers over the non-labeled ones implies that the diffusive fluxes of all labeled N

tracers follow the gradient of the non-labeled (or bulk) N tracers. In reality, diffusive fluxes of matter from N tracers from different sources may be opposed. However, numerical diffusion is known to be higher than background diffusion. Therefore, we consider the impact on the results to be small.

In summary, the model and N tracing applied here provide a useful basis for a meaningful analysis of our research questions.

## 4.2 Hypoxia under current environmental conditions

Previous observational studies of the hypoxic region off the Changjiang River Estuary have concluded that the regions north and south of 30 °N differ with respect to the nutrient sources contributing to hypoxia formation (Chi et al., 2017; Li et al., 2002; Zhu et al., 2011). The northern region is considered to be dominated by Changjiang River inputs, while oceanic nutrient sources are thought to be important in the southern region. Here, we present the first quantification of the relative contributions of the different nutrient sources to hypoxia formation. Our results show that the riverine contribution to GOC during the hypoxia

season (July to November) during 2008–2013 steadily decreases from the northwest to the southeast in the region typically affected by hypoxia (see Fig. 3). The riverine contribution to GOC dominates the northern region, supporting $78.0 \pm 5.9\%$ of GOC ($73.9 \pm 5.4\%$ attributed to Changjiang River), while oceanic N supports only $22.0 \pm 5.9\%$ ($15.2 \pm 3.7\%$ from Kuroshio; see Fig. 4a and Table S2) confirming that Changjiang River nutrient loads control $O_2$ consumption and thus hypoxia formation in the northern region.

When considering July–November averages in the southern region, the relative contribution of riverine N to GOC is $61.6 \pm 5.7\%$ ($56.9 \pm 5.1\%$ from Changjiang River), while the oceanic contribution is $38.4 \pm 5.7\%$, nearly twice the oceanic contribution in the northern region, with roughly equal contributions from Kuroshio and Taiwan Strait (see Fig. 4b and Table S2). However, analysis of the seasonal cycles of the source-specific contributions to GOC reveals that hypoxia expands whenever the Changjiang contribution to GOC increases (see Fig. 5 and Table S3). This demonstrates that the Changjiang

River also is the major factor for hypoxia development in the southern region.

High Changjiang contributions to GOC correspond to high FW thicknesses and thus strong stratification (see Fig. S5), the other essential factor for hypoxia formation and maintenance than biological $O_2$ consumption. In contrast, the hypoxic area is smaller during periods with high oceanic contributions as a result of reduced GOC and weaker stratification (see Figs. 5 and S5). Nevertheless, the relatively low subsurface $O_2$ concentrations in the oceanic water masses precondition the southern region for hypoxia formation.

Our analyses of the seasonal cycles of GOC and stratification in relation to the large-scale meridional winds further show that the East Asian monsoon and its year-to-year variability result in seasonal and year-to-year variability in the contributions to GOC, in stratification, and thus in hypoxia in the southern region (see Fig. 5). During winter monsoon (September–April), the southward winds transport the Changjiang River plume towards the southern region, where it arrives in August/September supporting the formation of hypoxia. The opposite occurs during summer monsoon (May–August), when the northward winds transport oceanic water masses into the southern region and push the river plume northward out of the southern region.

With respect to year-to-year variability, our analysis further shows that anomalies in the meridional winds during summer and during the transition from summer to winter monsoon (August–October) significantly affect the water mass distribution, and thus GOC and hypoxia on subseasonal scales, especially south of 30 °N (see Fig. 5 and Table S3). Particularly weak southward winds in September/October of 2008 resulted in a longer presence of the Changjiang River plume in the southern region, increasing stratification and GOC, which in turn caused the formation of the largest hypoxic area in that region during 2008–2013. The opposite occurred in July/August of 2013, when particularly strong northward winds pushed the Changjiang River plume north. This resulted in a drop in GOC and stratification, and in the vanishing of hypoxia, which had started forming in June through early July as anomalously weak northward winds allowed a southward transport of the Changjiang River plume. This is in agreement with Zhang et al. (2018) who simulated similar almost immediate responses of hypoxia to changes in the wind field as a result of the redistribution of Changjiang FW causing changes in stratification. Our study expands on their work by additionally demonstrating the importance of the redistribution of Changjiang nutrients for GOC.

These short-term changes in GOC, stratification and hypoxic area in response to variations in the large-scale winds illustrate the complexity of the region with respect to atmosphere–ocean interaction and their effect on hypoxia. They further highlight the necessity of a high spatio-temporal resolution of both model approaches and observations to understand the causes of hypoxia formation and maintenance in the ECS under current environmental conditions.

In general, our results are in agreement with Zhang et al. (2019) who found that riverine N supports 56% of water column integrated PP (8-year average) in the ECS regions shallower than 50 m, which are most susceptible to hypoxia (see Fig. 3). This confirms the high contribution of Changjiang N to GOC found for our analysis regions. Similarly, the comparably low oceanic contributions, especially in the northern region, are in line with their results. Zhang et al. (2019) also found a relatively high contribution (22%) of atmospheric N to PP suggesting that atmospheric N deposition should be considered in future biogeochemical modeling studies of the ECS.

### 4.3 Mitigation of hypoxia in the present and future

Our analysis of the changes in hypoxic area and hypoxic exposure under reduced Changjiang River N loads (see Fig. 6 and
Table 1) suggests a high potential of riverine N load reductions to mitigate hypoxia. The simulated average reduction in
hypoxic area by 56.5% is almost proportional to the N load reduction of 50% but varies significantly from year to year (39.9%
in 2009 to 91.4% in 2011). Hypoxia vanishes almost entirely during months of small hypoxic areas (May to August 2008,
September/October 2013; see Fig. S7) and maximum hypoxic areas during the seasonal cycle are significantly reduced in
response to decreased GOC supported by Changjiang River N inputs. However, comprehensive further studies are needed to
assess the effect of Changjiang River nutrient reductions on hypoxia. In this context, it is important to consider all processes
relevant for hypoxia formation. Zhou et al. (2017b) conducted a scenario with 50% lower Changjiang River nutrient loads and
found a reduction in hypoxic area by only about 20%. However, they did not consider SOC, a major sink of $O_2$ in the ECS
(Zhang et al., 2017), which may explain this discrepancy.

Analysis of the scenario where open-ocean $O_2$ concentrations were 20% lower than at present shows that the reduced lateral
supply due to declining open-ocean $O_2$ concentrations (Bopp et al., 2017) may significantly exacerbate hypoxia in the ECS,
indicated by the expansion of both hypoxic and anoxic areas by about 50% (see Fig. 6 and Table 1). However, we further found
that a 50% reduction of the Changjiang River N loads more than compensates for this increase in hypoxia.

The 20% reduction of open-ocean $O_2$ concentrations that was applied here corresponds to changes in subsurface $O_2$ of
about 30–40 $\mathrm{mmol\,O_2\,m^{-3}}$ projected by earth system models (Bopp et al., 2017), but does not consider other climate change
effects such as local changes in water temperature or stratification. For instance, an increase in temperature would reduce $O_2$
solubility, which would worsen $O_2$ conditions further as shown, e.g. for the NGoM (Laurent et al., 2018). Therefore, more
comprehensive studies on climate change impacts on hypoxia in the ECS are required.

### 4.4 Comparison with the northern Gulf of Mexico

In the following, we compare the hypoxic zones of the ECS and the NGoM with respect to their sizes and timing, based on
observations, and the main sources of $O_2$ consumption, based on the results of this study and an analogous nutrient tracing
study for the NGoM (Große et al., 2019). The key messages of this comparison are summarized in Table 2.

The spatial extent of observed hypoxia in the ECS is similar to that in the NGoM, affecting areas on the order of 15,000 $\mathrm{km^2}$.
In the NGoM, hypoxia is frequently observed from May to September when seasonal stratification is strongest due to high river-
ine freshwater inputs. In the ECS, the hypoxic season starts about one month later in June, with hypoxia forming directly off
the Changjiang River Estuary. Hypoxia can be observed until October/November, though the hypoxic zone moves southward
over summer.

In the NGoM, riverine nutrients from the Mississippi and Atchafalaya Rivers dominate $O_2$ consumption. In the ECS, only
in the region north of 30 °N is $O_2$ consumption dominated by riverine nutrient inputs that almost exclusively originate from
the Changjiang River. South of 30 °N, the oceanic contribution to $O_2$ consumption is comparable to that of riverine nutrients.
However, driven by year-to-year variability in the East Asian monsoon, the Changjiang River appears to be the main control

**Table 2.** Comparison of hypoxia in the northern Gulf of Mexico (NGoM) and the East China Sea (ECS).

| Characteristic | NGoM | ECS |
|---|---|---|
| Hypoxic area ($km^2$) | 15,000 [1] | 15,000 [2] |
| Hypoxic season | May–September [1] | June–November [3] |
| GOC controls (seasonal average) | riverine $\gg$ oceanic | >30 °N: riverine $\gg$ oceanic<br><30 °N: riverine $\approx$ oceanic |
| Main riverine source(s) | Mississippi & Atchafalaya | Changjiang |
| Controls of year-to-year variability | Mississippi & wind field | Changjiang & wind field |

[1] Rabalais et al. (2002); [2] Li et al. (2002), Zhu et al. (2011, 2017); [3] Wang et al. (2012)

of hypoxia formation also in the southern region. Similarly, the Mississippi River in concert with the local wind field seems to be the main control of year-to-year variability in hypoxic area in the NGoM. There, episodic upwelling-favorable winds can result in an offshore transport of Mississippi FW and nutrients leading to a reduction in stratification and $O_2$ consumption and thus hypoxia (Feng et al., 2014).

## 5 Conclusions

To our knowledge, this is the first study quantifying the relative contributions of individual riverine and oceanic nutrient sources on hypoxia formation in ECS using active element tracing. Therefore, it constitutes an important milestone towards the quantification of the contributions of riverine vs. oceanic nutrient sources to hypoxia formation in the ECS.

Our results suggest that N from the Changjiang River is the dominant driver of $O_2$ consumption ($73.9 \pm 5.4\%$) north of 30 °N under present-day environmental conditions (2008–2013). Contrary to observational insights and despite high contributions of N from Kuroshio and Taiwan Strait to $O_2$ consumption on seasonal time scales ($19.5 \pm 2.2\%$ and $18.9 \pm 3.2\%$, respectively), the Changjiang River waters are also the main factor for hypoxia formation south of 30 °N.

Our analysis highlights the importance of considering subseasonal time scales for understanding the controls of hypoxia formation and its year-to-year variability in this region. The East Asian monsoon and its associated change in large-scale wind patterns control water mass distribution and thus $O_2$ consumption and stratification. Year-to-year variability in the intensity of the winds can lead to significant differences in the amount of Changjiang River water in the southern region explaining year-to-year variability in hypoxia.

Reductions in the Changjiang River nutrient loads appear to have a high potential for mitigating hypoxia and to counteract the likely future decline of open-ocean $O_2$ supplied to the region. However, more comprehensive studies of both the effect of riverine nutrient load reductions and climate change effects are required.

*Code and data availability.* A modified version of the ROMS source code that facilitates writing of model diagnostics needed by the element tracing software (ETRAC) is available at https://github.com/FabianGrosse/ROMS_3.7_for_ETRAC.

ETRAC is available via https://github.com/FabianGrosse/ETRAC. Due to the large data size, ROMS and ETRAC output is freely available upon request from the corresponding author (fabian.grosse@dal.ca).

380 *Author contributions.* FG and KF conceived the study. FG set up the element tracing and conducted the analyses. HZ set up the model. AL helped with validation. FG wrote the manuscript with input from all co-authors.

*Competing interests.* The authors declare that they have no conflict of interest.

*Acknowledgements.* All figures in this article were created with MATLAB, benefiting significantly from the toolboxes for TEOS-10 (Mc-Dougall and Barker, 2011), the *cmocean* color schemes (Thyng et al., 2016) and *m_map* (https://www.eoas.ubc.ca/~rich/map.html). The
385 authors of these toolboxes are thankfully acknowledged. The geographical information used for creation of Fig. 1 were obtained from the Global Self-consistent, Hierarchical, High-resolution Geography Database (GSHHG; https://www.soest.hawaii.edu/wessel/gshhg/). We thank Compute Canada for providing computing resources under the resource allocation project qqh-593-ac. KF acknowledges funding from the Natural Sciences and Engineering Research Council of Canada (NSERC) Discovery program. Financial support to HZ from the China Scholarship Council (CSC) is gratefully acknowledged. We thank Hagen Radtke and one anonymous reviewer for their constructive criticism,
390 which helped to improve this manuscript.

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
