# Peer review of "Quantifying the contributions of riverine vs. oceanic nitrogen to hypoxia in the East China Sea"

_Biogeosciences, 2019_

## Referee Comment (RC1) · Hagen Radtke (Referee) · 8 Oct 2019

The article has a clear scientific objective and is written clearly and concisely. It provides new insights into East China Sea hypoxia.

Unfortunately I see two major shortcomings which need to be clarified before the article can be published. The first one is about the appropriateness of using a simplification of the nutrient tagging method for this application. The second one is about insufficient model validation. If these can be fixed and the authors show that (a) the method is applicable and (b) the model has a sufficient quality in the parameters in question, I would recommend publication of the article.

Please also note the supplement to this comment:
https://www.biogeosciences-discuss.net/bg-2019-342/bg-2019-342-RC1-supplement.pdf

**Supplement:**

**General remarks**

The article has a clear scientific objective and is written clearly and concisely. It provides new insights into East China Sea hypoxia.

Unfortunately I see two major shortcomings which need to be clarified before the article can be published. The first one is about the appropriateness of using a simplification of the nutrient tagging method for this application. The second one is about insufficient model validation. If these can be fixed and the authors show that (a) the method is applicable and (b) the model has a sufficient quality in the parameters in question, I would recommend publication of the article.

**Choice of the tagging method**

I see a serious issue with the applied tagging method.

**Problem**

Your equation (2) presented in line 101 does not describe the element tracing method described in Menesguen et al. (2006) and Radtke et al. (2012),

$$\frac{\partial C_X^i}{\partial t} \;\; = \;\; \nabla \cdot \left( \overline{\overline{D}} \nabla C_X^i \right) - \nabla \cdot \left( C_X^i \mathbf{v} \right) + R_{C_X} \cdot \frac{C_{X_{con}}^i}{C_{X_{con}}} \; . \tag{1}$$

but you rather use a simplification

$$\frac{\partial C_X^i}{\partial t} \;\; = \;\; \nabla \cdot \left( \overline{\overline{D}} \nabla C_X \right) \cdot \frac{C_X^i}{C_X} - \nabla \cdot \left( C_X \mathbf{v} \right) \cdot \frac{C_X^i}{C_X} + R_{C_X} \cdot \frac{C_{X_{con}}^i}{C_{X_{con}}} \; , \tag{2}$$

which deviates from the full equation.

In your 2017 publication ("A Novel Modeling Approach to Quantify the Influence of Nitrogen Inputs on the Oxygen Dynamics of the North Sea") you discussed this problem in a specific paragraph, but you do not state this difference here.

The effect is that in the simplified equation, mixing or advection of the tagged element is always driven by a gradient in the **total** concentration $C_X$. So your formulation does not allow a diffusion or advection of a tagged element $C_X^i$ against the gradient of the total concentration $C_X$.

This is especially problematic in this application, where you try to investigate how much oceanic N enters the (N-richer) coastal area. Your simplification prevents this transport. In this way, the contribution of a "local" source is systematically overestimated while that of a remote source is underestimated. I cannot see why it can be ruled out that this methodological error actually determines the result of your study.

**Suggested solution**

I suggest a simple experiment to quantify the impact of the simplification. In a first step, you initialize three passive tracers with the concentrations of

- $p_1$ = riverine N,

- $p_2$ = non-riverine N and

- $p_3$ = total N

at some single time step. Then you run the model for a few years and see how these spread. I expect that your numerical scheme will be linear and p1+p2=p3 will be maintained as it should.

Then, initialize two "active elements" (whose spreading is calculated by equation (2)):

- $a_1$ = riverine N,

- $a_2$ = non-riverine N.

with the same initial concentrations, using $p_3$ as their "parent element". Make them practically passive by setting $R_{p_1} = R_{p_2} = 0$. If the simplification error is negligible, $a_1$ should behave very similar to $p_1$ and $a_2$ to $p_2$, and you should end up with very similar ratios of non-riverine N to total N in both methods. Then you could present this as a verification that your simplification error is small.

If my expectation is right and the results will show a significant difference, this would mean that you have to apply the full rather than the simplified method for this application.

**Missing validation**

You refer to an existing publication for the model validation. That is not sufficient. Your study relies on the assumption that at least the following is reproduced by the model:

1. lateral nitrogen transport,

2. oxygen consumption rates,

3. hypoxic area extent.

You should then present model validation that proves that the model is capable to do that. I am thinking of

1. DIN observations,

2. benthic chamber lander O2 fluxes or, if not existent, at least primary production rates as a proxy,

3. observational-based estimates of the hypoxic area.

**Specific comments**

- L38: The correct reference for the element tracing method is Menesguen et al. 2006: "A new numerical technique for tracking chemical species in a multisource, coastal ecosystem applied to nitrogen causing Ulva blooms in the Bay of Brest (France)". In the Menesguen and Hoch 1997 paper, a more general method for tracking multiplicative properties of model state variables was described which only later in the later paper was applied for element tracing.

- Figure 1: I suggest to change the color scale. Firstly, it guides the reader's focus to the location of the shelf edge only and makes it hard to distinguish topographic features in the tracing region. Secondly, a scale like that is typically used the opposite way, having the darkest shades of blue at the deepest locations, I would recommend to stick to this habit to make it more intuitive for the reader.

- L61-62: Instantaneous benthic remineralization is a good choice if (a) sediment biogeochemistry is in a dynamic steady state (carbon accumulation negligible) and (b) the area is so deep that lateral transport of resuspended organic matter does not play a role. Both assumptions seem questionable here, please discuss the possible implications on your model results.

- L75: Please specify which rivers you prescribed, maybe by adding them as dots in Figure 1 or by supplying a table with their mouths' coordinates in the online supplement.

- L86: Please state earlier than in the "Discussion" section what motivates the reduced-oxygen scenarios and why you choose a 20% reduction.

- L122: The TN concentrations are actually monthly, daily values are only obtained by interpolation, correct? Please also change the caption of Figure 2.

- Table 1: How is "anoxic area" defined?

**Technical corrections**

- L59: Citation style is wrong here, please use the "citep" command if the reference can be omitted without changing the meaning of the sentence.

- L138: A comma is missing after "South of $32°N$".

- L278: A comma is missing after "e.g."

---

## Referee Comment (RC2) · Anonymous Referee #2 · 24 Oct 2019

This manuscript quantified the contribution of nitrogen from Changjiang and open ocean (Taiwan Strait and Kuroshio) to the hypoxia formation in the East China Sea and proposed the reduction of nitrogen from river as an efficient way to avoid hypoxia. In general, I can follow this manuscript. However, I also found many points needed to clarify before I can recommend its publication.

General comments

1. Do you include the particle organic nitrogen from rivers? On line 61, you mention only dissolved organic matter (DON) but show TN in Fig. 2. If your TN includes particle organic nitrogen, how did you determine the proportion of PON, DON and DIN (NO3

and NH4) in your input data of TN?

2. Consumption of oxygen by sediment is an important factor affects formation of hypoxia. What is your sediment condition? There is only one sentence (line 62) saying it but it is not enough.

3. You mention the importance of winds in the interannual variations. However, the change of wind speed in Fig. 5 is very small (<2 m/s?). Would you like to present more evidences for the processes related to winds? For example, you mentioned changes in flow field and turbulence but did not show any figures for these changes.

4. You emphasized the importance of Changjiang in this study. However, you actually did not consider the interannual variations in the Taiwan Strait and Kuroshio region because you used a nudging to climatology there. The same thing also occurs for the nitrogen from Yellow Sea. Therefore, your conclusion is not fair.

5. What is background for reduction of O2 in the open ocean by 20%? It is better for you to check the papers for DO change at 137E line for some evidences.

6. I did not find figures showing interannual and seasonal variations in spatial variations of bottom DO concentration from your model. Apparently, they are important to your model validation because you can find some observations showing such figures. Without a serious validation of model results, no people in China can follow your suggestion on reduction of nitrogen input by 50%.

Specific comments

Line 29-31: This statement is not correct.

Line 74: please use full spell for 'FW'.

Line 84-86: ". . .the initial and open-boundary O2 concentrations were reduced by 20% throughout the water column in regions deeper than 200 m. . ." How much O2 reduction from the Kuroshio boundary or Taiwan Strait boundary?

Line 110: "...Minjiang, Hanjiang and Oujiang Rivers; grouped into one source..." You mean Hanjiang River or Qiantangjiang River? In Figure 1, Hanjiang River is not inside the tracing region. How did you trace the N of it?

Line 112: What is your evidence for that the tracer cannot reenter the tracing region?

Line115: "...To spin up the tracing, we first re-ran year 2006 three times. For the first iteration, all N mass already in the system was attributed to the small rivers...." What's the purpose of doing this?

Line125: Figure2. In 2009, 2011, 2013, the Changjiang discharge and TN concentration seem to have the similar trend, but 2010 and 2012 the opposite. Why does this happen?

Line 135: do you have any data to verify the GOC given here?

Supplement: what is your purpose to show PEA/D not PEA itself?

---

## Author Comment (AC1) · 30 Oct 2019

Dear reviewers, dear editor,

We would like to thank the two reviewers for the generally positive feedback and constructive criticism on our manuscript.

Below, you can find all comments made (in black) and our individual responses (in green). We hope our responses and suggested changes satisfy the reviewers.

Kind regards,
Fabian Große

On behalf of all authors

**Review #1 by Hagen Radtke**

**General remarks**
The article has a clear scientific objective and is written clearly and concisely. It provides new insights into East China Sea hypoxia.
Unfortunately I see two major shortcomings which need to be clarified before the article can be published. The first one is about the appropriateness of using a simplification of the nutrient tagging method for this application. The second one is about insufficient model validation. If these can be fixed and the authors show that (a) the method is applicable and (b) the model has a sufficient quality in the parameters in question, I would recommend publication of the article.

**Choice of the tagging method**
I see a serious issue with the applied tagging method.
**Problem**
Your equation (2) presented in line 101 does not describe the element tracing method described in Menesguen et al. (2006) and Radtke et al. (2012), which deviates from the full equation.

[equations removed]

In your 2017 publication ("A Novel Modeling Approach to Quantify the Influence of Nitrogen Inputs on the Oxygen Dynamics of the North Sea") you discussed this problem in a specific paragraph, but you do not state this difference here.
The effect is that in the simplified equation, mixing or advection of the tagged element is always driven by a gradient in the total concentration Cx. So your formulation does not allow a diffusion or advection of a tagged element Cx_i against the gradient of the total concentration Cx_i.
This is especially problematic in this application, where you try to investigate how much oceanic N enters the (N-richer) coastal area. Your simplification prevents this transport. In this way, the contribution of a "local" source is systematically overestimated while that of a remote source is underestimated. I cannot see why it can be ruled out that this methodological error actually determines the result of your study.
**Suggested solution**

I suggest a simple experiment to quantify the impact of the simplification. In a first step, you initialize three passive tracers with the concentrations of
• p1 = riverine N,
• p2 = non-riverine N and
• p3 = totalN
at some single time step. Then you run the model for a few years and see how these spread. I expect that your numerical scheme will be linear and p1+p2=p3 will be maintained as it should.
Then, initialize two "active elements" (whose spreading is calculated by equation (2)):
• a1 = riverine N,
• a2 = non-riverine N.
with the same initial concentrations, using p3 as their "parent element". Make them practically passive by setting R_p1 = R_p2 = 0. If the simplification error is negligible, a1 should behave very similar to p1 and a2 to p2, and you should end up with very similar ratios of non-riverine N to total N in both methods. Then you could present this as a verification that your simplification error is small.
If my expectation is right and the results will show a significant difference, this would mean that you have to apply the full rather than the simplified method for this application.

**Reply:** Thank you for spotting this. In fact, the advection term in our Eq. (2) was written incorrectly and confused the differential term describing the change in the concentration at a location with the discretization of the calculation of the advective transports across the grid cell interfaces. We will correct this. Advection of a labelled tracer is calculated correctly, yet your statement on diffusion holds. However, we are aware of this and think it is reasonable to assume that the numerical diffusion from the applied first-order upstream advection scheme (MPDATA) is much larger than the turbulent diffusion, and thus the effect of this simplification is small. Your comment made us aware of a limitation of our treatment of advection (= multiplication of transport flux of unlabelled tracer with relative fraction of labelled tracer). Specifically, this approach only works correctly for advection schemes based on absolute concentrations, as we have used, but would yield incorrect results for gradient based advection schemes. We will include a statement on this in the revised manuscript.

**Missing validation**
You refer to an existing publication for the model validation. That is not sufficient. Your study relies on the assumption that at least the following is reproduced by the model:
1. lateral nitrogen transport, 2. oxygen consumption rates, 3. hypoxic area extent.
You should then present model validation that proves that the model is capable to do that. I am thinking of
   1. DIN observations,
   2. benthic chamber lander O2 fluxes or, if not existent, at least primary production rates as a proxy,
   3. observational-based estimates of the hypoxic area.

**Reply:** Publicly available DIN and oxygen observations for the coastal areas of the East China Sea are sparse. However, we will include a validation of the spatial patterns of DIN (or nitrate) based on the available observations. Observation-based estimates of hypoxic area for individual years are also available from the literature and we will include these numbers in

the revised manuscript. A qualitative comparison of simulated and observed bottom $O_2$ concentrations (both spatially and temporally) is provided by Zhang et al. (https://www.biogeosciences-discuss.net/bg-2019-341/; Fig. 3). A brief comparison of simulated sediment $O_2$ consumption with literature values is provided in the discussion of the companion paper by Zhang et al.

**Specific comments**

L38: The correct reference for the element tracing method is Menesguen et al. 2006: "A new numerical technique for tracking chemical species in a multisource, coastal ecosystem applied to nitrogen causing Ulva blooms in the Bay of Brest (France)". In the Menesguen and Hoch 1997 paper, a more general method for tracking multiplicative properties of model state variables was described which only later in the later paper was applied for element tracing.

**Reply:** One could argue that the element tracing is only a special case of the general method described in Ménesguen & Hoch (1997), but we will change the reference as suggested.

Figure 1: I suggest to change the color scale. Firstly, it guides the reader's focus to the location of the shelf edge only and makes it hard to distinguish topographic features in the tracing region. Secondly, a scale like that is typically used the opposite way, having the darkest shades of blue at the deepest locations, I would recommend to stick to this habit to make it more intuitive for the reader.

**Reply:** We will change the color scale as suggested (i.e., invert it).

L61-62: Instantaneous benthic remineralization is a good choice if (a) sediment biogeochemistry is in a dynamic steady state (carbon accumulation negligible) and (b) the area is so deep that lateral transport of resuspended organic matter does not play a role. Both assumptions seem questionable here, please discuss the possible implications on your model results.

**Reply:** Indeed, Song et al. (2016; https://doi.org/10.1016/j.dsr2.2015.04.012) determined (based on observations) that on average about 45% of the settled organic matter carbon (~14% of primary production) are permanently buried in the sediments of the East China Sea, although with quite some spatial variability. Consequently, simulated sediment $O_2$ consumption (SOC) may overestimate the observations, which is indicated when comparing simulated and observation-based SOC rates. This likely affects simulated near-bottom $O_2$, but we consider it having only a small effect on the relative contributions of individual sources to gross $O_2$ consumption (GOC), as this limitation equally applies to all labelled N sources.
Sediment resuspension may result in a lower riverine contribution near the river mouths, and higher contributions in more distant areas (vice versa for oceanic contribution). However, except for typhoon events, wind speed (and thus resuspension) is generally lower during summer. Song et al. also state that resuspension may particularly play a role in fall when wind speed starts increasing with the change in the monsoon cycle. We will include this in the discussion of the revised manuscript.

L75: Please specify which rivers you prescribed, maybe by adding them as dots in Figure 1 or by supplying a table with their mouths' coordinates in the online supplement.

**Reply:** We will provide a Table with the river names and mouth locations in the revised supplement in order to not overload Fig. 1.

L86: Please state earlier than in the "Discussion" section what motivates the reduced-oxygen scenarios and why you choose a 20% reduction.

**Reply:** Yes, we will include this in the scenario description in the Methods section of the revised manuscript.

L122: The TN concentrations are actually monthly, daily values are only obtained by interpolation, correct? Please also change the caption of Figure 2.

**Reply:** Yes, river load concentrations for the Changjiang River are monthly data from Global NEWS. Only freshwater discharge is daily. We will correct this in the caption.

Table 1: How is "anoxic area" defined?

**Reply:** Anoxic area is defined as the region experiencing $O_2$ concentrations of 0. We will add this in the text of the revised manuscript.

**Technical corrections**

L59: Citation style is wrong here, please use the "citep" command if the reference can be omitted without changing the meaning of the sentence.

**Reply:** Will be corrected.

L138: A comma is missing after "South of 32°N".

**Reply:** Will be corrected.

L278: A comma is missing after "e.g."

**Reply:** We consistently use no comma after "e.g." (like on the Biogeosciences website (https://www.biogeosciences.net/for_authors/manuscript_preparation.html; "English guidelines and house standards")

**Review #2 by anonymous referee**

This manuscript quantified the contribution of nitrogen from Changjiang and open ocean (Taiwan Strait and Kuroshio) to the hypoxia formation in the East China Sea and proposed the reduction of nitrogen from river as an efficient way to avoid hypoxia. In general, I can follow this manuscript. However, I also found many points needed to clarify before I can recommend its publication.

**General comments**

1. Do you include the particle organic nitrogen from rivers? On line 61, you mention only dissolved organic matter (DON) but show TN in Fig. 2. If your TN includes particle organic

nitrogen, how did you determine the proportion of PON, DON and DIN (NO3 and NH4) in your input data of TN?

**Reply:** Yes, the river forcing includes information for small and large detritus (=PON), DON, NO3 and NH4 (=DIN). However, actual forcing data is only available for NO3 and NH4 (from Global NEWS). For the 3 groups of PON/DON constant concentrations were applied. We will include those in the description of the river forcing in the Methods section.

2. Consumption of oxygen by sediment is an important factor affects formation of hypoxia. What is your sediment condition? There is only one sentence (line 62) saying it but it is not enough.

**Reply:** The statement on line 62 implies that all organic material that sinks to the seafloor is remineralized immediately, with a fraction of 75% being lost to dinitrogen via benthic denitrification (Fennel et al., 2006; https://doi.org/10.1029/2005GB002456). We will clarify this in the description of the biogeochemical model and include a paragraph on potential implications of this relatively simple approach in the discussion of the revised manuscript.

3. You mention the importance of winds in the interannual variations. However, the change of wind speed in Fig. 5 is very small (<2 m/s?). Would you like to present more evidences for the processes related to winds? For example, you mentioned changes in flow field and turbulence but did not show any figures for these changes.

**Reply:** In terms of absolute numbers, the year-to-year differences in wind speed are indeed relatively small (1-3 m/s). However, considering the discussed events, e.g. September 2008 vs. 2013 and June 2013, it can be seen that the relative change is quite significant as absolute wind speed does not exceed 4 m/s (during these events). In the supplement (Fig. S2), we provided time series of monthly averaged potential energy anomaly (PEA; a measure for water column stability), which implicitly reflects changes in turbulence as vertical mixing is reduced under more stable conditions. This is discussed on lines 180-188. Along with the PEA time series, we show time series of freshwater thickness associated with the Changjiang River discharge. We use freshwater thickness is a measure of the total amount of Changjiang freshwater in the top 25 m of the water column. Changes in freshwater thickness can only result from changes in lateral transport and in discharge from the Changjiang. However, the discharge is quite similar in the first half of 2008 and 2013 (see Fig. 2), thus differences in freshwater thickness between the two years need to be due to differences in transport of freshwater from the Changjiang to the southern analysis region. In addition, changes in freshwater thickness and PEA (Fig. S2) clearly coincide with anomalous wind events (Fig. 2). We think the effect on stratification/turbulence is sufficiently addressed by the PEA time series. However, we will consider including an example for the effect of wind on surface currents for one or two of the discussed events in the supplement (similar to Fig. S1).
Is this sufficient?

4. You emphasized the importance of Changjiang in this study. However, you actually did not consider the interannual variations in the Taiwan Strait and Kuroshio region because you used a nudging to climatology there. The same thing also occurs for the nitrogen from Yellow Sea. Therefore, your conclusion is not fair.

**Reply:** It is correct, that we do not fully resolve interannual variations in the nitrogen supply from the oceanic sources due to the nudging of nitrate concentrations to a climatology. However, the nutrient supply from the Kuroshio occurs primarily in the subsurface, with open-ocean subsurface concentrations showing significantly lower absolute values (e.g. Liu et al., 2016; http://dx.doi.org/10.1016/j.jmarsys.2015.05.010) and lower variability than coastal waters with river influence. Therefore, variations in volume transport of Kuroshio intrusions are likely the main cause for interannual variations in nutrient supply from the Kuroshio. These are resolved by the model. Similarly, nitrogen concentrations in Taiwan Strait (e.g. Chen et al., 2004; https://doi.org/10.1016/j.marchem.2004.01.006) are significantly lower than in the river inputs (by factor 10 to 100). Therefore, we consider the effect of interannual variability in nitrogen levels in the oceanic sources small compared to the variability in the river loads. We will include this in the discussion of the revised manuscript.

5. What is background for reduction of O2 in the open ocean by 20%? It is better for you to check the papers for DO change at 137E line for some evidences.

**Reply:** This 20% reduction corresponds to the reduction in subsurface $O_2$ levels in the northeast Pacific projected by Earth System models (Bopp et al., 2017). We are particularly interested in potential future changes in the $O_2$ conditions off the Changjiang. We therefore base our scenario on these future projections rather than observations of past changes. As suggested by reviewer #1, we will clarify this already in the scenario description in the Methods section.

6. I did not find figures showing interannual and seasonal variations in spatial variations of bottom DO concentration from your model. Apparently, they are important to your model validation because you can find some observations showing such figures. Without a serious validation of model results, no people in China can follow your suggestion on reduction of nitrogen input by 50%.

**Reply:** This model-data comparison is provided in the companion paper of Zhang et al. (https://www.biogeosciences-discuss.net/bg-2019-341/; Fig. 3). We consider it redundant providing the same analysis in this manuscript. However, we will expand the discussion of the revised manuscript with respect to model agreement with observations and explicitly refer to this companion paper at the appropriate locations.
We further like to stress that the 50% reduction scenario is only a single model realization and does not suffice to make actual recommendations. We ran this scenario to obtain first insight into how the system may respond to nitrogen load reductions. The 50% reduction was chosen in analogy to Zhou et al. (2017; https://doi.org/10.1016/j.marchem.2017.07.006) who did not consider sediment $O_2$ consumption in their model, thus missing relevant parts of the system. This is also discussed on lines 267-270.
From our point of view, the strongest statement with respect to the potential impact of river load reductions on hypoxia reads as follows in the original version of the manuscript:
"Our analysis of the changes in hypoxic area and hypoxic exposure under reduced Changjiang River N loads (see Fig. 6 and Table 1) underlines the high potential of riverine N load reductions to mitigate hypoxia." (lines 263-264)
We will rephrase it to:

"Our analysis of the changes in hypoxic area and hypoxic exposure under reduced Changjiang River N loads (see Fig. 6 and Table 1) **suggests a** high potential of riverine N load reductions to mitigate hypoxia."

As we only ran a single reduction scenario, we are fully aware that we are not in the position of making an actual recommendation and we do not mean to be prescriptive in any way.

**Specific comments**

Line 29-31: This statement is not correct.

**Reply:** We will rephrase the last part of the sentence to: "with strong southwestward winds in winter and weak northwestward winds in summer supporting stronger northward water mass transport in summer than in winter."

Line 74: please use full spell for 'FW'.

**Reply:** "FW" is first introduced on line 20. After that we consistently use "FW" instead of "freshwater", which we would like to keep.

Line 84-86: ". . .the initial and open-boundary O2 concentrations were reduced by 20% throughout the water column in regions deeper than 200 m. . ." How much O2 reduction from the Kuroshio boundary or Taiwan Strait boundary?

**Reply:** We only reduced the $O_2$ levels at the open boundaries of the model domain (see Fig. 1 in the manuscript) in regions deeper than 200 m. We will clarify this.

Line 110: ". . .Minjiang, Hanjiang and Oujiang Rivers; grouped into one source. . ." You mean Hanjiang River or Qiantangjiang River? In Figure 1, Hanjiang River is not inside the tracing region. How did you trace the N of it?

**Reply:** We accidentally put a wrong river name, it has to be "Qiantangjiang River", which will be corrected.

Line 112: What is your evidence for that the tracer cannot reenter the tracing region?

**Reply:** This is owed to the tracing setup, which does not keep track of the origin of a tracer once it leaves the tracing region. In reality, nutrients could be recirculated into/re-enter the region. We will clarify this.

Line115: "… To spin up the tracing, we first re-ran year 2006 three times. For the first iteration, all N mass already in the system was attributed to the small rivers." What's the purpose of doing this?

**Reply:** This is done to spin up the model (it is common practice to do so). Note that we do not have information on the actual distributions of nitrogen from the different sources in the region. Therefore, we have to start from an arbitrary distribution for which all nitrogen tracers are attributed to the small rivers (any other of the traced sources would be equally good). We then run the tracing multiple times (3 times in this case) with the same forcing until we

achieve a statistical steady state meaning that the distributions of tracers associated with the different sources do not change between December 31 of two subsequent iterations. At this point the model is considered as spun up. This way we make sure that our results are not affected by the arbitrary initial distributions. This is also stated in lines 117-119, but we will try to make this more clear.

Line125: Figure2. In 2009, 2011, 2013, the Changjiang discharge and TN concentration seem to have the similar trend, but 2010 and 2012 the opposite. Why does this happen?

**Reply:** This is a good question, to which we don't have a sure answer. To some extent, this could be a result of combining information from two different sources (Global NEWS for nitrogen concentrations, Datong gauge measurements for discharge). However, more likely this relates to the strong river floods in 2010 and 2012 (indicated by the much higher discharge peaks in both years compared to the other years). However, we could not find literature explaining this in more detail and it is outside of our field of expertise to answer this question (and outside of the intended scope of this manuscript).

Line 135: do you have any data to verify the GOC given here? Supplement: what is your purpose to show PEA/D not PEA itself?

**Reply:** We do not have data for GOC but we will include a comparison of simulated sediment $O_2$ consumption with observation-based estimates in the discussion.
PEA increases with increasing water depth, which would give stronger weight to deeper regions within the analysis regions. To avoid this, we show PEA/D accounting for this spatial variability of water depth.

---

## Author Response (AR2)

Dear Editor,

We would like to thank the two Reviewers for their assessment of our revised manuscript. We are pleased that Reviewer 1 is satisfied with this updated version.

Below, you can find all comments by Reviewer 2 (in black), our individual responses and changes made to the manuscript (both in green; line numbers refer to revised submission).

We hope you will find our responses and changes satisfactory.

Kind regards, Fabian Große

On behalf of all authors

**Review by Anonymous Reviewer**

Thank the authors for their effort on improving this manuscript. I carefully read the response notes and revised manuscript and agree that the authors clarified some points in this revision. However, there are still some unclear and even wrong points. Following are my notes as I read revised manuscript (line number is from the manuscript with modification marked).

1. (line 20): should be m^3 yr^-1. Reply: Thank you for spotting this; we have corrected it.

2. (lines 29-32): The authors cited Bian et al. (2013) to relate seasonal variation of Kuroshio intrusion and monsoon winds. However, this relation is wrong. Please read carefully the papers by Yang et al. (2011, 2012) and Guo et al. (2006). The authors cited these papers but failed to understand their conclusion on seasonal variation in Kuroshio intrusion: change in density field is important while that in local wind is not as important as the authors described here (actually, not only here). The authors cited this relation many times in this manuscript). Following papers are also helpful for the authors to understand Kuroshio water intrusion into the East China Sea. Please note the Kuroshio intrusion is strong from autumn to winter but weak in summer.

Yang, et al. (2018). Topographic beta spiral and onshore intrusion of the Kuroshio Current. Geophysical Research Letters, 45, 287–296.

Oey, L. Y., Hsin, Y. C., & Wu, C. R. (2010). Why does the Kuroshio northeast of Taiwan shift shelf ward in winter? Ocean Dynamics, 60(2), 413–426.

**Reply:** Our description was indeed a bit unprecise and could be misunderstood. As pointed out by the reviewer, Kuroshio intrusions are stronger in winter than in summer (e.g. Bian et al., 2013; Guo et al., 2006). However, due to the changes in wind field with the monsoon cycle, northward water mass transport of Kuroshio subsurface waters on the East China Sea shelf is stronger in summer than in winter (see Fig. 4 in Guo et al., 2006), which implies that—even though Kuroshio intrusions are weaker—more Kuroshio water can reach 'our' southern analysis region. Yang et al. (2012) explicitly link the northward flow of the intruded Kuroshio

waters on the shelf to the northeastward winds and the related offshore Ekman transport and formation of a pressure gradient during the summer monsoon. Hence, there is a clear connection between large scale wind field (i.e. monsoon phase) and water mass transport on the shelf. However, we rephrased this text passage to make it less misleading (lines 29-32).

3. (line 61): Does the model include oceanic dissolved organic matter? If not, why? If the influence from rivers includes DON's effect but that from open ocean does not, the comparison for role of river and oceanic water is unfair.

**Reply:** Oceanic DON is considered to be part of the small and suspended detritus pool (SDet). The rationale for including an additional river-related model state variable is that riverine organic matter is more refractory than organic matter produced in the marine environment (i.e. by primary production), which is accounted for by the difference in remineralization rates (by one order of magnitude) between the riverine DON (which remineralizes slowly) and marine small and suspended detritus (SDet). We added a statement to clarify this (lines 62-65).

**4. (line 66): what is evidence for 75%?**

**Reply:** Equation (15) in Fennel et al. (2006; https://doi.org/10.1029/2005GB002456) (and corresponding text/supplement) describes in detail that the oxidation of 1 mole of organic matter (containing 16 moles of nitrogen) yields 4 moles of ammonium (NH4+) and 6 moles of dinitrogen (N2). Thus, 12 out of 16 N atoms (= 75%) are lost via benthic denitrification. We included the reference to Fennel et al. (2006) at the end of the sentence (line 67).

**5. (line 67): what is evidence for 115:4?**

**Reply:** Appendix A in Fennel et al. (2013; https://doi.org/10.1002/jgrc.20077) provides the detailed derivation of this relationship. We included the reference to Fennel et al. (2013) at the end of the sentence (lines 68/69).

6. (line 76-77): what is physical background for 7 and 10 days? Why should they be different? **Reply:** There is no reason for them to be different. These happened to be the values that were used in the simulations carried out, but we confirmed that the results are not sensitive to the exact value. In fact, they are almost indistinguishable if the times scales are varied between 1 week and 2 weeks. We added a corresponding statement to the manuscript (lines 78–80).

**7. (line 80): not 11 rivers.**

**Reply:** There are 11 rivers inside the entire model domain. However, later on in the manuscript, we only consider those inside the element tracing region, which are 4 in total incl. the Changjiang River. We did not apply any changes to the text.

8. (line 81-83): I cannot understand the relation between each state variable here and TN given in Fig. 2.

**Reply:** The total nitrogen (TN) river load (Fig. 2) is calculated as the sum of all nitrogen state variables discharged by the Changjiang River (i.e. NO3, NH4, large and small detritus, DON, phyto- and zooplankton). We added this to the text (lines 142/143) and to caption of Fig. 2. We also added how phytoplankton and zooplankton loads are prescribed (line 89) and that NO3 concentrations dominate the riverine TN concentrations, while the detritus and dissolved organic matter contribution varies between 8% and 13% (lines 145/146).

**9. (line 85): what are evidences for these values?**

**Reply:** As stated in the manuscript, these values are conservatively assumed as data are not publicly available. The time series of the TN concentration in Fig. 2 shows that the TN concentration in the Changjiang River ranges between 115 and 180 mmol m-3, which is dominated by the nitrate concentration. Hence, organic inputs account for only 8-13% of all N inputs in our model and the effect of these values on the overall result can be considered small. We did not apply any changes to the text.

10. (line 90-91): what is background for reduction in only N load. Since you also include phosphate as state variable, reduction in only N load will change N/P ratio in the region affected by the Changjiang River in the model calculation, which is however not a realistic situation because phosphate load also changes if N load changes.

**Reply:** Riverine N is derived mostly from diffuse agricultural sources (through use of industrial fertilizer). For P, there is no such comparable source and, as a result, agricultural sources of P are comparably small. Furthermore, different studies document that N is delivered in great access to P, with reported N:P molar ratios of ranging between 22.5 (Tong et al., 2017; their Fig. 4; https://doi.org/10.1016/j.jhazmat.2016.09.011) up to 66 (Fennel and Testa, 2019; their Table 1; https://doi.org/10.1146/annurev-marine-010318-095138). Therefore, the reduction of N only is a sensible scenario and reflects an assumption of reduced industrial fertilizer use in agriculture. For clarification, we added this reference to industrial fertilizer use

11. (line 95): the same question is also for 20% O2 reduction case. What is reason for O2 reduction in a warming situation? If it is the biogeochemical process, reduction in O2 concentration and increasing in nutrient concentration occur together. Therefore, it is not reasonable to reduce only O2 concentration but keep the same nutrient concentration in oceanic water.

**Reply:** The projected changes in open-ocean oxygen concentrations result from both increased temperature (i.e. reduced solubility) and changes in biological activity (Bopp et al., 2017; https://doi.org/10.1098/rsta.2016.0323), primarily due to increased turnover rates. This means that concentrations of the different N pools (e.g. nitrate and ammonium vs. detritus) may differ, but it does not imply that overall N levels would change. Our scenario is meant to illustrate potential changes in ECS hypoxia only in response to reduced open-ocean oxygen levels, which we emphasize at the appropriate locations in the manuscript. Hence, we did not apply any changes to the text.

12. (Line 118): what is the exact way to diagnose the N tracer? Does this means that you did not solve the equation for each source of nitrate (Equation 2))?

**Reply:** Equation (2) is solved for each source by multiplication of the bulk flux (e.g. total nitrate uptake during primary production) with the relative fraction of a state variable from a specific source (e.g. nitrate containing N from the Changjiang) of the corresponding bulk state variable (e.g. nitrate). We added a clarifying statement (lines 122-124).

13. (line 118): I am wondering the separation here will cause some artificial problem in the area where the phosphate, not nitrate, is a limiting element for phytoplankton growth. Can

the authors add some words to address this concern? As we know, the offshore area of Changjiang River has a very high N/P ratio.

**Reply:** The separation between different N sources is not affected by the regional differences in N vs. P limitation. This is implied by the calculation of the source-specific fluxes as a product of the overall fluxes (which are accounting for potential P limitation) with the source-specific fraction of the consumed state variable. We hope and believe this is now clarified by the statement added to address the previous comment and, therefore, did not apply any additional changes to the text.

14. (Figure 5): As I comment before, the anomaly of winds speed is not very large. The authors emphasize the positive anomaly of wind speed on August 2013 when there is no contribution of Changjiang River. However, this positive anomaly plus the climatology on August 2013 is not larger than wind speed on July 2013. However, we can find contribution of Changjiang River on July 2013 in this figure. Therefore, disappear of contribution from Changjiang River is not necessarily caused by local winds. Another question for Fig. 5 is what is the area for averaging the meridional wind speed.

**Reply:** As stated in our response to a similar comment on the first version of the manuscript, the absolute anomalies in meridional wind speed may not be very large, but the relative anomalies are significant. The differences between the July and August 2013 need to be considered in the context of the prior development as a result of the anomaly wind (and thus the currents) in the region in June. The anomalously weak northward winds in June enable the Changjiang River plume to reach the analysis region (different to 2008, when the steady northward winds in May/June push the plume farther north). The subsequently consistent northward winds in July/August push the Changjiang waters even farther north, resulting in a steady decline of the Changjiang contribution in the region. In other words, the starting conditions for July and August 2013 are quite different as a result of the wind fields and resulting water mass distributions prior to each month. We did not apply any changes to the text.

The averaging area for the wind is provided in the caption of Fig. 5.

15. (line 189-190): Again, I have to say that the relation between wind and movement of Kuroshio and Taiwan Strait water given here is wrong.

**Reply:** As outlined in our response to comment #2 and as shown in Fig. S1 (supplement) and confirmed by, e.g. Guo et al. (2006), northward water mass transport on the shelf is stronger in summer than in winter driven by the northeastward winds during summer monsoon (Yang et al., 2012). As a result, the oceanic contributions from Taiwan Strait and Kuroshio increase in the analysis region. Again, we'd like to emphasize that this does not apply to the strength of Kuroshio intrusions northeast of Taiwan themselves (as pointed out by the reviewer and clarified in relation to comment #2), but for the water mass transport in general. As the water masses in the southern shelf area are dominated by oceanic water masses, this northward transport causes an increase in the oceanic contributions. We added the reference to Yang et al. (2012) to the text (line 193) but do not explain in detail the connection between wind forcing and water mass transport as this is not the focus of the paper and has been done previously by others.

16. (line 221): I would like to suggest the authors to present the same Fig. 5 for this case (50% reduction of N load). The comparison between two cases is very helpful to understand the role of Changjiang source of nitrate.

**Reply:** Thank you for this suggestion. We decided to add this figure to the supplement (new Fig. S7) to not overload the main text. We included text passages referring to this figure in the Results section (lines 227-231) and in the Discussion (lines 327-329).

17. (line 250): resuspension in this area is largely attributed to strong tidal currents. The authors can easily confirm this from satellite images for suspended matter (e.g., Shi et al., 2011). There is always high concentration of suspended matter in this area although the wind wave is not always high. In addition, Shi et al. (2011) also demonstrated a strong neap and spring tidal cycle in suspended matter in this area.

Shi, W., M. Wang, and L. Jiang (2011), Spring-neap tidal effects on satellite ocean color observations in the Bohai Sea, Yellow Sea, and East China Sea, J. Geophys. Res., 116, C12032, doi:10.1029/2011JC007234.

**Reply:** Thank you for pointing us to the paper by Shi et al. This is indeed a valid point, yet, beyond the scope of this manuscript in our point of view. We, therefore, rephrased the paragraph on resuspension, now stating that it is relevant and recommending its inclusion in future studies (lines 256-259).

**Quantifying the contributions of riverine vs. oceanic nitrogen to hypoxia in the East China Sea**

Fabian Große1,2, Katja Fennel1, Haiyan Zhang1,3, and Arnaud Laurent1

1Department of Oceanography, Dalhousie University, Halifax, NS, Canada

[revised manuscript text omitted]

---

## Author Response (AR3)

Dear Reviewers, dear Editor,

We would like to thank again the two Reviewers for the generally positive feedback and constructive criticism on our manuscript.

Below, you can find all comments (in black), our individual responses and changes made to the manuscript (both in green). We hope our responses and changes satisfy the Reviewers.

Kind regards, Fabian Große

On behalf of all authors

**Review #1 by Hagen Radtke**

**General remarks**

The article has a clear scientific objective and is written clearly and concisely. It provides new insights into East China Sea hypoxia.

Unfortunately I see two major shortcomings which need to be clarified before the article can be published. The first one is about the appropriateness of using a simplification of the nutrient tagging method for this application. The second one is about insufficient model validation. If these can be fixed and the authors show that (a) the method is applicable and (b) the model has a sufficient quality in the parameters in question, I would recommend publication of the article.

**Reply**: Thank you for your generally positive assessment. We believe that the two major points that you raised are addressed in the revision. Please see detailed responses below.

**Choice of the tagging method**

I see a serious issue with the applied tagging method.

**Problem**

Your equation (2) presented in line 101 does not describe the element tracing method described in Menesguen et al. (2006) and Radtke et al. (2012), which deviates from the full equation.

**[equations removed]**

In your 2017 publication ("A Novel Modeling Approach to Quantify the Influence of Nitrogen Inputs on the Oxygen Dynamics of the North Sea") you discussed this problem in a specific paragraph, but you do not state this difference here.

The effect is that in the simplified equation, mixing or advection of the tagged element is always driven by a gradient in the total concentration Cx. So your formulation does not allow a diffusion or advection of a tagged element Cx\_i against the gradient of the total concentration Cx\_i.

This is especially problematic in this application, where you try to investigate how much oceanic N enters the (N-richer) coastal area. Your simplification prevents this transport. In this way, the contribution of a "local" source is systematically overestimated while that of a

remote source is underestimated. I cannot see why it can be ruled out that this methodological error actually determines the result of your study.

**Suggested solution**

I suggest a simple experiment to quantify the impact of the simplification. In a first step, you initialize three passive tracers with the concentrations of

- p1 = riverine N,
- p2 = non-riverine N and
- p3 = totalN

at some single time step. Then you run the model for a few years and see how these spread. I expect that your numerical scheme will be linear and p1+p2=p3 will be maintained as it should.

Then, initialize two "active elements" (whose spreading is calculated by equation (2)):

- a1 = riverine N,
- a2 = non-riverine N.

with the same initial concentrations, using p3 as their "parent element". Make them practically passive by setting  $R_p1 = R_p2 = 0$ . If the simplification error is negligible, a1 should behave very similar to p1 and a2 to p2, and you should end up with very similar ratios of non-riverine N to total N in both methods. Then you could present this as a verification that your simplification error is small.

If my expectation is right and the results will show a significant difference, this would mean that you have to apply the full rather than the simplified method for this application.

**Reply:** Thank you for spotting this. The advection term in our Eq. (2) was written incorrectly and confused the differential term describing the change in the concentration at a location with the discretization of the calculation of the advective transports across the grid cell interfaces. Advection of a labelled tracer is calculated correctly and we corrected Eq. (2) accordingly.

In fact, not only the advection term but also the diffusion term had to be corrected as its previous formulation in Eq. (2) implied that diffusion of a labeled tracer is calculated as the change in concentration of the corresponding non-labeled (i.e. bulk) tracer and the ratio of labelled over non-labeled tracer at location (x,y,z). However, the discretized diffusion flux across each individual grid cell interface is based on the gradient of the bulk tracer across that interface and the ratio of labeled over non-labeled tracer in the source cell (i.e. the cell in which the bulk tracer concentration is reduced by the diffusion flux across an individual interface).

While we are aware of the point you made that diffusion fluxes of labeled tracers (across individual grid cell interfaces) follow the gradient of the bulk tracer in our approach, we know that numerical diffusion in the MPDATA advection scheme here applied is much higher than turbulent diffusion. Thus, the effect of our simplification is small. This aspect is discussed in the new section 4.1 of the revised manuscript (lines 255-260).

**Missing validation**

You refer to an existing publication for the model validation. That is not sufficient. Your study relies on the assumption that at least the following is reproduced by the model:

1. lateral nitrogen transport, 2. oxygen consumption rates, 3. hypoxic area extent.

You should then present model validation that proves that the model is capable to do that. I am thinking of

- 1. DIN observations,
- 2. benthic chamber lander O2 fluxes or, if not existent, at least primary production rates as a proxy,
- 3. observational-based estimates of the hypoxic area.

**Reply:** We have now included a validation of surface and bottom dissolved inorganic nitrogen (DIN) based on observational data from Gao et al. (2015) in the supplement (Text S3, Figs. S2-S4). A reference to Figs. S2-S4 was included in the model description (line 84) and a discussion of model skill with respect to DIN, sediment oxygen consumption and hypoxic area has been added to the discussion section (part of new Sect. 4.1; lines 227-240). We did not include additional validation of hypoxic area as we consider this redundant with the validation provided in the companion paper by Zhang et al. in this Special Issue (https://www.biogeosciences-discuss.net/bg-2019-341/). We refer to this paper where appropriate in the new discussion section 4.1 of the revised manuscript (lines 225, 231).

**Specific comments**

L38: The correct reference for the element tracing method is Menesguen et al. 2006: "A new numerical technique for tracking chemical species in a multisource, coastal ecosystem applied to nitrogen causing Ulva blooms in the Bay of Brest (France)". In the Menesguen and Hoch 1997 paper, a more general method for tracking multiplicative properties of model state variables was described which only later in the later paper was applied for element tracing.

**Reply: Changed as suggested.**

Figure 1: I suggest to change the color scale. Firstly, it guides the reader's focus to the location of the shelf edge only and makes it hard to distinguish topographic features in the tracing region. Secondly, a scale like that is typically used the opposite way, having the darkest shades of blue at the deepest locations, I would recommend to stick to this habit to make it more intuitive for the reader.

**Reply: We inverted the color scale in Fig. 1.**

L61-62: Instantaneous benthic remineralization is a good choice if (a) sediment biogeochemistry is in a dynamic steady state (carbon accumulation negligible) and (b) the area is so deep that lateral transport of resuspended organic matter does not play a role. Both assumptions seem questionable here, please discuss the possible implications on your model results.

**Reply:** Indeed, Song et al. (2016; https://doi.org/10.1016/j.dsr2.2015.04.012) determined (based on observations) that on average about 45% of the settled organic matter carbon (~14% of primary production) are permanently buried in the sediments of the East China Sea, although with quite some spatial variability. Consequently, simulated sediment  $O_2$  consumption (SOC) may overestimate the observations, which is consistent with our comparison of simulated and observation-based SOC rates. This likely affects simulated near-bottom  $O_2$ , but we consider it having only a small effect on the relative contributions of

individual sources to gross  $O_2$  consumption (GOC), as this limitation equally applies to all labelled N sources.

Sediment resuspension may result in a lower riverine contribution near the river mouths, and higher contributions in more distant areas (vice versa for oceanic contribution). However, except for typhoon events, wind speed (and thus resuspension) is generally lower during summer. Song et al. also state that resuspension may particularly play a role in fall when wind speed starts increasing with the change in the monsoon cycle. We will include this in the discussion of the revised manuscript.

We included this discussion in the revised manuscript (part of new Sect. 4.1, lines 236-246).

L75: Please specify which rivers you prescribed, maybe by adding them as dots in Figure 1 or by supplying a table with their mouths' coordinates in the online supplement.

**Reply:** We added a table with geographical locations of river input cells to the revised supplement (Table S1). This table is referred to at the appropriate locations in the revised main text: lines 77 and 117.

L86: Please state earlier than in the "Discussion" section what motivates the reduced-oxygen scenarios and why you choose a 20% reduction.

**Reply:** We added a sentence citing the projection by Earth System Models (Bopp et al., 2017) as the motivation for this reduction at the end of Sect. 2.1 (lines 90-94) of the revised manuscript.

L122: The TN concentrations are actually monthly, daily values are only obtained by interpolation, correct? Please also change the caption of Figure 2.

**Reply:** Yes, river load concentrations for the Changjiang River are monthly data from Global NEWS. Only freshwater discharge is daily. We corrected the figure caption in the revised manuscript; it's been correct in the main text.

Table 1: How is "anoxic area" defined?

**Reply:** Anoxic area is defined as the region experiencing  $O_2$  concentrations of 0. We added our definition of anoxic area to the caption of Table 1.

**Technical corrections**

L59: Citation style is wrong here, please use the "citep" command if the reference can be omitted without changing the meaning of the sentence.

**Reply: Has been corrected.**

L138: A comma is missing after "South of 32°N".

Reply: Has been corrected.

L278: A comma is missing after "e.g."

**Reply:** We consistently use no comma after "e.g." (like on the Biogeosciences website (https://www.biogeosciences.net/for authors/manuscript preparation.html; "English guidelines and house standards"). Hence, we did not change it.

**Review #2 by anonymous referee**

This manuscript quantified the contribution of nitrogen from Changjiang and open ocean (Taiwan Strait and Kuroshio) to the hypoxia formation in the East China Sea and proposed the reduction of nitrogen from river as an efficient way to avoid hypoxia. In general, I can follow this manuscript. However, I also found many points needed to clarify before I can recommend its publication.

**General comments**

1. Do you include the particle organic nitrogen from rivers? On line 61, you mention only dissolved organic matter (DON) but show TN in Fig. 2. If your TN includes particle organic nitrogen, how did you determine the proportion of PON, DON and DIN (NO3 and NH4) in your input data of TN?

**Reply:** Yes, the river forcing includes information for small and large detritus (=PON), DON, NO3 and NH4 (=DIN). However, actual forcing data is only available for NO3 and NH4 (from Global NEWS). Therefore, constant concentrations were applied to small and large detritus and DON for all river inputs. We included these values in the revised model description (lines 81/82).

2. Consumption of oxygen by sediment is an important factor affects formation of hypoxia. What is your sediment condition? There is only one sentence (line 62) saying it but it is not enough.

**Reply:** The statement on line 62 implies that all organic material that sinks to the seafloor is remineralized immediately, with a fraction of 75% being lost to dinitrogen via benthic denitrification (Fennel et al., 2006; https://doi.org/10.1029/2005GB002456). Sediment  $O_2$  consumption is proportional to the release of ammonium from benthic remineralization. We expanded the description of the benthic remineralization in the revised model description (lines 61-65) and added a discussion comparing simulated benthic fluxes with literature values and describing its potential impacts on the model results (part of new section 4.1, lines 236-240).

3. You mention the importance of winds in the interannual variations. However, the change of wind speed in Fig. 5 is very small (<2 m/s?). Would you like to present more evidences for the processes related to winds? For example, you mentioned changes in flow field and turbulence but did not show any figures for these changes.

**Reply:** In terms of absolute numbers, the year-to-year differences in wind speed are indeed relatively small (1-3 m/s). However, considering the discussed events, e.g. September 2008 vs. 2013 and June 2013, it can be seen that the relative change is quite significant as absolute wind speed does not exceed 4 m/s (during these events). In the supplement (Fig. S5;

previously Fig. S2), we provide time series of monthly averaged potential energy anomaly (PEA; a measure for water column stability) over water depth D, which implicitly reflects changes in turbulence as vertical mixing is reduced under more stable conditions. This is discussed on lines X-Y. Along with the PEA/D time series, we show time series of freshwater thickness associated with the Changjiang River discharge. We use freshwater thickness as a measure of the total amount of Changjiang freshwater in the top 25 m of the water column. Changes in freshwater thickness can only result from changes in lateral transport and in discharge from the Changjiang. However, the discharge is quite similar in the first half of 2008 and 2013 (see Fig. 2), thus differences in freshwater thickness between the two years need to be due to differences in transport of freshwater from the Changjiang to the southern analysis region. In addition, changes in freshwater thickness and PEA/D (Fig. S5) clearly coincide with anomalous wind events (Fig. 5). We believe that this PEA/D time series addresses the Reviewer's point about the effect on stratification/turbulence.

Furthermore, we added Sect. S6 and Fig. S6 to the revised supplement, illustrating the differences in the surface current fields during June and October of 2008 and 2013 in relation to the year-to-year differences in the wind field (Fig. 5) during these two months. The figure shows how strength and direction of the coastal current are affected by the differences in the wind field between both years. We added a reference to Fig. S6 on lines 190/191 and 195-198 of the revised main text.

4. You emphasized the importance of Changjiang in this study. However, you actually did not consider the interannual variations in the Taiwan Strait and Kuroshio region because you used a nudging to climatology there. The same thing also occurs for the nitrogen from Yellow Sea. Therefore, your conclusion is not fair.

**Reply:** It is correct, that we do not fully resolve interannual variations in the nitrogen supply from the oceanic sources due to the nudging of nitrate concentrations to a climatology. However, the nutrient supply from the Kuroshio occurs primarily in the subsurface, with open-ocean subsurface concentrations showing significantly lower absolute values (e.g. Liu et al., 2016; http://dx.doi.org/10.1016/j.jmarsys.2015.05.010) and lower variability than coastal waters with river influence. Therefore, variations in volume transport of Kuroshio intrusions are likely the main cause for interannual variations in nutrient supply from the Kuroshio. These are resolved by the model. Similarly, nitrogen concentrations in Taiwan Strait (e.g. Chen et al., 2004; https://doi.org/10.1016/j.marchem.2004.01.006) are significantly lower than in the river inputs (by factor 10 to 100). Therefore, we consider the effect of interannual variability in nitrogen levels in the oceanic sources small compared to the variability in the river loads.

We added this discussion to the new discussion section on model validation and study limitations (Sect. 4.1, lines 247-254)

5. What is background for reduction of O2 in the open ocean by 20%? It is better for you to check the papers for DO change at 137E line for some evidences.

**Reply:** This 20% reduction corresponds to the reduction in subsurface  $O_2$  levels in the northeast Pacific projected by Earth System models (Bopp et al., 2017). We are particularly interested in potential future changes in the  $O_2$  conditions off the Changjiang. We therefore base our scenario on these future projections rather than observations of past changes.

**We included the motivation for the 20% reduction of open-ocean $O_2$ in the revised scenario description (Sect. 2.1, lines 90-94)**

6. I did not find figures showing interannual and seasonal variations in spatial variations of bottom DO concentration from your model. Apparently, they are important to your model validation because you can find some observations showing such figures. Without a serious validation of model results, no people in China can follow your suggestion on reduction of nitrogen input by 50%.

**Reply:** This model-data comparison is provided in the companion paper of Zhang et al. in this same Special Issue (https://www.biogeosciences-discuss.net/bg-2019-341/; Fig. 3). We consider it redundant providing the same analysis in this manuscript.

However, we included a qualitative discussion of model-data/-literature agreement of observed hypoxic areas in the new discussion section 4.1 of the revised manuscript.

We like to stress that the 50% reduction scenario is only a single model realization and does not suffice to make actual recommendations. We ran this scenario to obtain a first insight into how the system may respond to nitrogen load reductions. The 50% reduction was chosen in analogy to Zhou et al. (2017; https://doi.org/10.1016/j.marchem.2017.07.006) who did not consider sediment O2 consumption in their model, thus missing relevant parts of the system. This is discussed on lines X-Y (previously lines 267-270). As we only ran a single reduction scenario, we are fully aware that we are not in the position of making an actual recommendation and we do not mean to be prescriptive in any way.

Our statement about the potential impact of river load reductions on hypoxia reads as follows in the original version of the manuscript:

"Our analysis of the changes in hypoxic area and hypoxic exposure under reduced Changjiang River N loads (see Fig. 6 and Table 1) underlines the high potential of riverine N load reductions to mitigate hypoxia." (lines 263-264)

We rephrased it to (change indicated in bold):

"Our analysis of the changes in hypoxic area and hypoxic exposure under reduced Changjiang River N loads (see Fig. 6 and Table 1) **suggests a** high potential of riverine N load reductions to mitigate hypoxia." (lines 312-313)

**Specific comments**

Line 29-31: This statement is not correct.

**Reply:** We rephrased the last part of the sentence to: "with strong southwestward winds in winter and weak northwestward winds in summer supporting stronger northward water mass transport in summer than in winter." (lines 29-31)

Line 74: please use full spell for 'FW'.

**Reply:** "FW" is first introduced on line 20. After that we consistently use "FW" instead of "freshwater". Therefore, we did not change it.

Line 84-86: ". . .the initial and open-boundary O2 concentrations were reduced by 20% throughout the water column in regions deeper than 200 m. . ." How much O2 reduction from the Kuroshio boundary or Taiwan Strait boundary?

**Reply:** We only reduced the  $O_2$  levels at the open boundaries of the model domain (see Fig. 1 in the manuscript) in regions deeper than 200 m. This means that the inflow of  $O_2$  into the model domain (not the tracing domain) is reduced by 20%. Changes in the  $O_2$  transports in the interior of the model domain (incl. the Kuroshio boundary) can deviate from this reduction due to the internal dynamics.

We expanded the description on what the change in the open-boundary conditions means for the  $O_2$  transports into the model domain in the revised manuscript (Section 2.1, lines 90-99).

Line 110: "...Minjiang, Hanjiang and Oujiang Rivers; grouped into one source..." You mean Hanjiang River or Qiantangjiang River? In Figure 1, Hanjiang River is not inside the tracing region. How did you trace the N of it?

**Reply:** We accidentally put a wrong river name, it has to be "Qiantangjiang River", which we corrected in the revised manuscript (lines 116/117).

Line 112: What is your evidence for that the tracer cannot reenter the tracing region?

**Reply:** This is owed to the tracing setup, which does not keep track of the origin of a tracer once it leaves the tracing region. In reality, nutrients could be recirculated into/re-enter the region.

We added text clarifying this to the revised methods section 2.2 (lines 118-120).

Line115: "... To spin up the tracing, we first re-ran year 2006 three times. For the first iteration, all N mass already in the system was attributed to the small rivers." What's the purpose of doing this?

**Reply:** This is done to spin up the model (it is common practice to do so). Note that we do not have information on the actual distributions of nitrogen from the different sources in the region. Therefore, we have to start from an arbitrary distribution for which all nitrogen tracers are attributed to the small rivers (any other of the traced sources would be equally good). We then run the tracing multiple times (3 times in this case) with the same forcing until we achieve a statistical steady state meaning that the distributions of tracers associated with the different sources do not change between December 31 of two subsequent iterations. At this point the model is considered as spun up. This way we make sure that our results are not affected by the arbitrary initial distributions.

We rewrote the last paragraph of Sect. 2.2 in the revised manuscript, hoping that it is more clear now (lines 122-128).

Line125: Figure 2. In 2009, 2011, 2013, the Changjiang discharge and TN concentration seem to have the similar trend, but 2010 and 2012 the opposite. Why does this happen?

**Reply:** This is a good question, to which we don't have a definitive answer to. We also note that this question is related to hydrology and thus is well outside the intended scope of the manuscript and our core area of expertise. If we had to speculate, we'd say that to some extent, this could be a result of combining information from two different sources (Global

NEWS for nitrogen concentrations, Datong gauge measurements for discharge). However, more likely this relates to the strong river floods in 2010 and 2012 (indicated by the much higher discharge peaks in both years compared to the other years). We could not find literature explaining this in more detail and emphasize again that it is outside of our field of expertise and outside of the intended scope of this manuscript.

Line 135: do you have any data to verify the GOC given here? Supplement: what is your purpose to show PEA/D not PEA itself?

**Reply:** We do not have data for GOC but we included a comparison of simulated sediment  $O_2$  consumption with observation-based estimates in the discussion (new Sect. 4.1, lines 229-232).

PEA increases with increasing water depth, which would give stronger weight to deeper regions within the analysis regions. To avoid this, we show PEA/D accounting for this spatial variability of water depth. We added a statement on this to the revised supplement section S5 (page 6, lines 17-20).

[revised manuscript text omitted]